# NOCA-1 functions with γ-tubulin and in parallel to Patronin to assemble non-centrosomal microtubule arrays in *C. elegans*

Shaohe Wang[1,2], Di Wu[1], Sophie Quintin[3,4], Rebecca A Green[1], Dhanya K Cheerambathur[1], Stacy D Ochoa[1], Arshad Desai[1], Karen Oegema[1]*

[1]Ludwig Institute for Cancer Research, Department of Cellular and Molecular Medicine, University of California, San Diego, La Jolla, United States; [2]Biomedical Sciences Graduate Program, University of California, San Diego, La Jolla, United States; [3]Institut Génétique Biologie Moléculaire Ceasllulaire, Faculté de médecine, Université de Strasbourg, Strasbourg, France; [4]Institut Clinique de la Souris, Illkirch-Graffenstaden, France

*For correspondence: koegema@ucsd.edu

Competing interests: The authors declare that no competing interests exist.

**Abstract** Non-centrosomal microtubule arrays assemble in differentiated tissues to perform mechanical and transport-based functions. In this study, we identify *Caenorhabditis elegans* NOCA-1 as a protein with homology to vertebrate ninein. NOCA-1 contributes to the assembly of non-centrosomal microtubule arrays in multiple tissues. In the larval epidermis, NOCA-1 functions redundantly with the minus end protection factor Patronin/PTRN-1 to assemble a circumferential microtubule array essential for worm growth and morphogenesis. Controlled degradation of a γ-tubulin complex subunit in this tissue revealed that γ-tubulin acts with NOCA-1 in parallel to Patronin/PTRN-1. In the germline, NOCA-1 and γ-tubulin co-localize at the cell surface, and inhibiting either leads to a microtubule assembly defect. γ-tubulin targets independently of NOCA-1, but NOCA-1 targeting requires γ-tubulin when a non-essential putatively palmitoylated cysteine is mutated. These results show that NOCA-1 acts with γ-tubulin to assemble non-centrosomal arrays in multiple tissues and highlight functional overlap between the ninein and Patronin protein families.

## Introduction

Differentiated cells assemble non-centrosomal microtubule arrays to perform structural, mechanical, and transport-based functions (*Keating and Borisy, 1999*; *Bartolini and Gundersen, 2006*). Examples include the neuronal microtubule arrays that structure axons and dendritic arbors (*Kuijpers and Hoogenraad, 2011*), longitudinal arrays of parallel microtubules in syncytial myotubes (*Warren, 1974*; *Tassin et al., 1985*), and non-centrosomal arrays in epithelial cells (*Keating and Borisy, 1999*; *Bartolini and Gundersen, 2006*). In simple epithelia, cells build arrays of parallel microtubules that run along their apical–basal axis (*Keating and Borisy, 1999*; *Bartolini and Gundersen, 2006*; *Brodu et al., 2010*; *Feldman and Priess, 2012*), whereas desmosomal cell–cell junctions organize microtubule arrays that form around the periphery of stratified epithelial cells in mouse skin (*Lechler and Fuchs, 2007*; *Sumigray et al., 2012*).

The radial organization of centrosomal arrays arises from the fact that microtubules are nucleated, and their nascent minus ends capped and anchored, by centrosomally targeted protein complexes. Similarly, assembly of non-centrosomal microtubule arrays is likely to involve targeting of microtubule nucleating, as well as minus-end protection and/or anchoring factors, to non-centrosomal sites.

**eLife digest** Microtubules are hollow, rigid filaments that are found in the cells of animals and other eukaryotes. These filaments are built from smaller building blocks called tubulin heterodimers; and in dividing animal cells, they mainly emerge from structures called centrosomes. When a cell is dividing, arrays of microtubules that originate from centrosomes help assemble the spindle-like structure that segregates the chromosomes.

Many non-dividing or specialized cells—including neurons, skin cells and muscle fibers—assemble other arrays of microtubules that do not emerge from centrosomes, but nevertheless perform a variety of structural, mechanical and transport-based roles. Compared to the centrosomal arrays, much less is known about how these non-centrosomal microtubules are assembled.

A vertebrate protein called 'ninein' had previously been shown to be involved in anchoring microtubules at centrosomes. Ninein can change its localization from centrosomes to the cell surface in mammalian skin cells, suggesting that it might also have a role in assembling the peripheral microtubule arrays that are found in these cells. Now, Wang et al. have identified a protein from worms called NOCA-1, which contains a region similar to the part of ninein that was previously shown to be needed to anchor microtubules at centrosomes.

The experiments show that NOCA-1 guides the assembly of non-centrosomal microtubule arrays in multiple tissues in *C. elegans* worms. This includes in the outer layer of the worm's larvae, which is similar to mammalian skin. The results also highlight that NOCA-1 performs many of the same roles as a member of the Patronin family of proteins called PTRN-1, which interacts with the 'minus' end of a microtubule to prevent the microtubule from breaking apart.

Wang et al. also found that NOCA-1 works with another protein called $\gamma$-tubulin, which helps new microtubules to form and also interacts with microtubule minus ends. In contrast, PTRN-1 works independently of $\gamma$-tubulin. This suggests that NOCA-1 works together with $\gamma$-tubulin to protect new microtubule ends or promote their assembly, a role similar to what has been proposed for Patronin family proteins. Overall, Wang et al.'s results highlight the importance of ninein-related proteins in the assembly of non-centrosomal microtubule arrays and suggest overlapping roles for the ninein and Patronin families of proteins.

Important current goals include identifying the factors that control the assembly of non-centrosomal arrays and determining the extent of overlap between the mechanisms utilized at centrosomes and non-centrosomal sites in different tissues.

Complexes containing $\gamma$-tubulin, a specialized tubulin isoform implicated in microtubule nucleation (*Zheng et al., 1995*; *Oegema et al., 1999*; *Kollman et al., 2011*), are thought to contribute to the assembly of both centrosomal and non-centrosomal arrays. During the differentiation of *Drosophila* tracheal epithelial cells, both $\gamma$-tubulin complexes, and the center of microtubule nucleation in regrowth experiments, transition from centrosomes to the apical cell surface (*Brodu et al., 2010*). In *Caenorhabditis elegans*, $\gamma$-tubulin is also targeted to the cell surface in the embryonic epidermis and germline, and the apical cell surface in the intestinal epithelium (*Zhou et al., 2009*; *Fridolfsson and Starr, 2010*; *Feldman and Priess, 2012*).

Ninein is a large coiled-coil protein that localizes to the sub-distal appendages of mother centrioles (*Mogensen et al., 2000*), where it is thought to anchor centrosomal microtubules (*Dammermann and Merdes, 2002*; *Delgehyr et al., 2005*). During the differentiation of mouse cochlear epithelial cells, ninein re-localizes from centrosomes to the apical surface (*Mogensen et al., 2000*; *Moss et al., 2007*); ninein re-localization also occurs during the differentiation of stratified epithelial cells in the mouse epidermis, where it targets to desmosomal junctions (*Lechler and Fuchs, 2007*). Inhibition of the core desmosomal component, desmoplakin, disrupts ninein targeting and formation of the peripheral non-centrosomal microtubule array (*Lechler and Fuchs, 2007*), but direct evidence that ninein is important for array formation is currently lacking.

The Patronin/CAMSAP/Nezha family of minus end-associated proteins, conserved among animals with differentiated tissues (*Baines et al., 2009*), are also implicated in the formation of non-centrosomal arrays (*Akhmanova and Hoogenraad, 2015*). Members of this protein family are thought to be involved in protecting microtubule minus ends from depolymerizing kinesins (*Goodwin and Vale, 2010*;

*Hendershott and Vale, 2014*; *Jiang et al., 2014*). *Drosophila* and *C. elegans* each have one family member (Patronin and PTRN-1, respectively), whereas vertebrates have three (calmodulin-regulated spectrin-associated protein or CAMSAP1-3). Although initially identified in cultured epithelial cells (*Meng et al., 2008*; *Jiang et al., 2014*), the main in vivo phenotypes associated with knockdown of Patronin/CAMSAP/Nezha family members have been in neurons (*Chuang et al., 2014*; *King et al., 2014*; *Marcette et al., 2014*; *Richardson et al., 2014*; *Yau et al., 2014*).

As outlined above, γ-tubulin and Patronin respectively harbor minus-end nucleation and protection activities, and ninein is proposed to anchor microtubules. Mechanistic work has also raised the possibility of functional redundancies between minus end-associated factors. For example, in addition to being a microtubule nucleator, γ-tubulin complexes can cap microtubule minus ends (*Keating and Borisy, 2000*; *Wiese and Zheng, 2000*). Similarly, CAMSAP-tubulin stretches may function as seeds that allow microtubule regrowth (*Tanaka et al., 2012*; *Jiang et al., 2014*), and both ninein and Patronin family members localize to junctional complexes (*Lechler and Fuchs, 2007*; *Meng et al., 2008*) where they could serve anchoring functions. Hence, another important open question is the extent to which minus end-associated factors function collaboratively or redundantly during microtubule array assembly in vivo.

Here, we characterize the *C. elegans* protein NOCA-1 (non-centrosomal array 1), a protein we identified in a prior high-content screen because its inhibition phenocopied the effect of γ-tubulin removal on germline morphology (*Green et al., 2011*). We show that NOCA-1 shares homology with vertebrate ninein and identify isoforms that are necessary and sufficient for NOCA-1 function in three different tissues. We explore the functional relationship between NOCA-1, γ-tubulin, and Patronin/PTRN-1 in the assembly of non-centrosomal microtubule arrays. In the larval epidermis, NOCA-1 functions with γ-tubulin in parallel to Patronin/PTRN-1 to assemble a circumferential microtubule array required for larval development. In the germline and embryonic epidermis, NOCA-1 functions independently of Patronin to promote assembly of microtubule arrays required for nuclear positioning. Cumulatively, our results suggest that NOCA-1 functions together with γ-tubulin to direct the assembly of non-centrosomal arrays in multiple tissues and highlight functional overlap between the ninein and Patronin families of microtubule cytoskeleton-controlling proteins.

## Results

### NOCA-1 has multiple isoforms with a shared C-terminal domain that is homologous to a region of vertebrate ninein

The *noca-1* locus is large (23 kb) and more complex than typical for *C. elegans* genes, encoding eight alternatively spliced isoforms that share a common 466 amino acid C-terminal domain with a predicted coiled-coil region (*Figure 1A*). Sequence homology searches identified similarity between this C-terminal domain of nematode NOCA-1 proteins and vertebrate nineins (*Figure 1A* and *Figure 1—figure supplement 1*). Ninein (*NIN*) and the related ninein-like protein (*NINL*) are homologous in their N- and C-termini but differ in their central region. The domain common to NOCA-1 isoforms is homologous to the ninein-specific central region that is absent in ninein-like protein (*Figure 1A* and *Figure 1—figure supplement 1*). This ninein-specific region resides within a larger domain suggested to be required for the microtubule anchoring function of centrosomal ninein (*Delgehyr et al., 2005*). We refer to the C-terminal domain of NOCA-1 common to all isoforms as the ninein homology domain (NHD).

NOCA-1 isoforms can be partitioned into two groups based on their sequence features: three short isoforms (d, e and g) that contain the NHD, and five long isoforms (a, b, c, f and h) that contain the NHD as well as an additional 205 shared amino acids that we will refer to as the Long Isoform Common Region (LICR). Each isoform also has a unique N-terminal extension (*Figure 1A*, *rainbow colors*) that varies in length from 18 to 251 amino acids. Thus, all NOCA-1 isoforms contain a common C-terminal domain with homology to the central ninein-specific region of vertebrate ninein.

### NOCA-1 and Patronin/PTRN-1 redundantly promote larval development and viability

To examine the in vivo functions of NOCA-1, we began by analyzing the phenotype of a *noca-1* deletion that affects all isoforms by removing the NHD (*ok3692*; *Figure 1A*). Immunoblotting with an

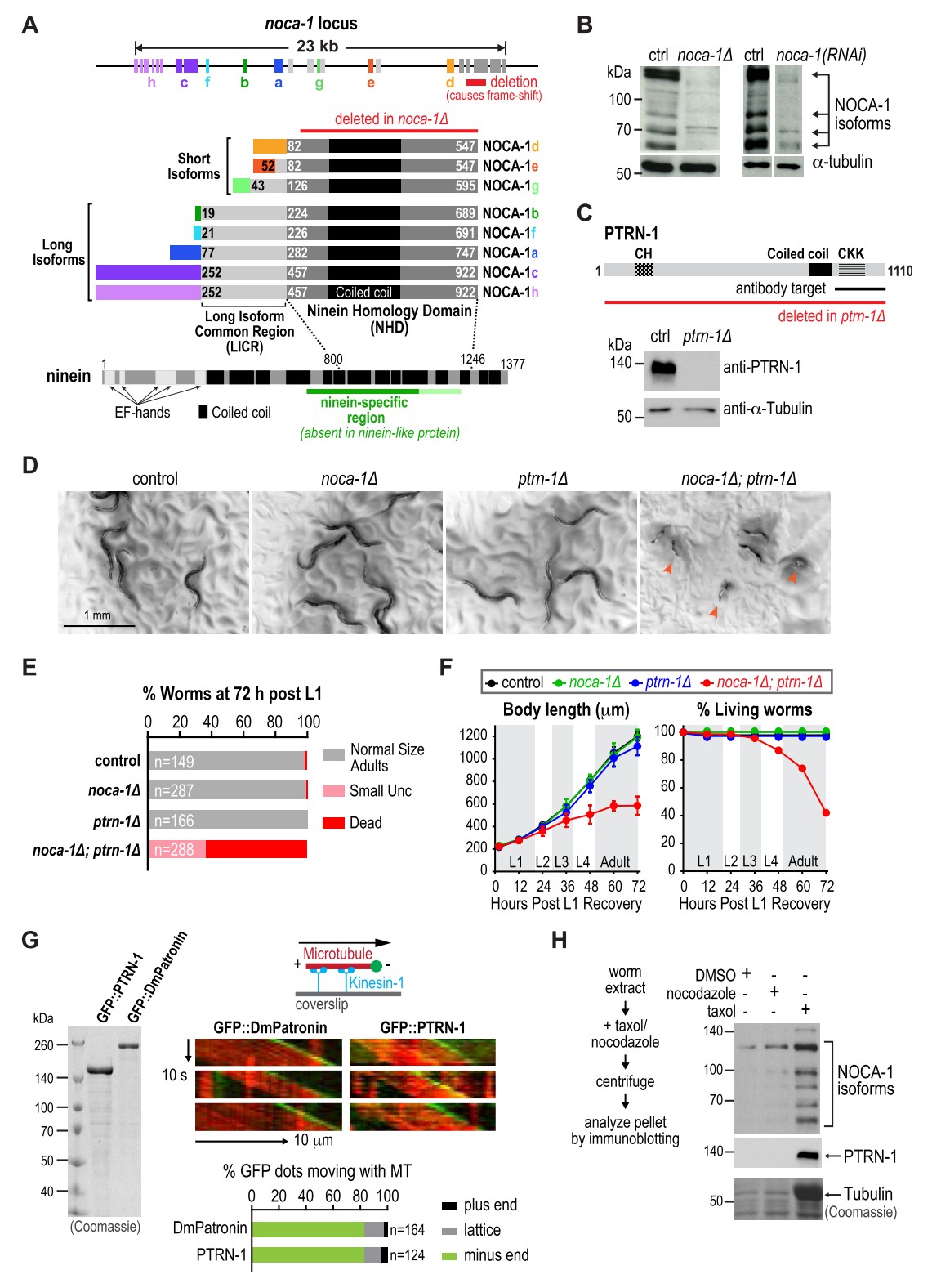

**Figure 1**. NOCA-1 is a protein with homology to vertebrate ninein that functions redundantly with PTRN-1/Patronin to promote larval development and viability. (**A**) Schematics of the *noca-1* locus, encoded NOCA-1 isoforms, and a short human ninein isoform showing the region with homology to NOCA-1 (alignment in *Figure 1—figure supplement 1A*). The region of ninein absent from (dark green) or with low homology to (light green) ninein-like protein is underlined. Red line above the NOCA-1 isoforms shows the region deleted in the ok3692 allele. (**B**) Immunoblot of NOCA-1 in lysates from control,

*Figure 1. continued on next page*

Figure 1. Continued

noca-1Δ, and noca-1(RNAi) worms. (**C**) Top: schematic of the *Caenorhabditis elegans* Patronin homolog, PTRN-1. Bottom: immunoblot of PTRN-1 in lysates from control and *ptrn-1Δ* worms. (**D**) Images of control and mutant worms 72 hr post L1 recovery (snapshots from *Video 1*). Arrowheads mark dead worms. (**E**) Plot of percentage of normal-sized adults, small uncs, and dead worms 72 hr post L1 for the indicated genotypes. n is number of worms analyzed in 3–5 independent experiments. (**F**) Plots of body length (*left*) and % living worms (*right*) vs time for worms with the indicated genotypes. (**G**) Left: Coomassie blue staining of recombinant proteins purified from baculovirus-infected insect cells. Right top: schematic of flow-cell-based kinesin gliding assay. Right center: kymographs showing microtubule gliding in the presence of indicated GFP-tagged proteins. Right bottom: plot of frequencies of plus end, minus end, or side binding. (**H**) Left: flow chart of microtubule co-sedimentation experiment. Right: immunoblots probing for NOCA-1 or PTRN-1 (top and center) and Coomassie blue staining showing tubulin (bottom) after sedimentation. Markers are in kDa. Coiled-coil predictions were performed using Paircoil2 (28 aa window, 0.025 threshold). Error bars are SEM.

The following figure supplements are available for figure 1:

**Figure supplement 1**. NOCA-1 has homology to vertebrate ninein.

**Figure supplement 2**. Expanded view of the immunoblot for NOCA-1 in lysates from control and *noca-1(RNAi)* worms shown in the right panel of *Figure 1B*.

**Figure supplement 3**. Construction of a deletion allele for the gene encoding *C. elegans* Patronin, PTRN-1.

**Figure supplement 4**. Hydrodynamic analysis of purified NOCA-1 and PTRN-1 proteins.

**Figure supplement 5**. Purified NOCA-1 binds to microtubules in aggregated forms.

antibody against the NHD coiled-coil recognized four major species that were absent or strongly reduced in extracts from *noca-1Δ* and *noca-1(RNAi)* worms (*Figure 1B* and *Figure 1—figure supplement 2*), indicating that at least four isoforms are expressed at detectable levels. Consistent with our prior work (*Green et al., 2011*), *noca-1Δ* worms were sterile and exhibited germline phenotypes equivalent to γ-tubulin depletion confirming that NOCA-1 has an essential role in assembly of the germline microtubule array. However, aside from germline abnormalities, *noca-1Δ* adult worms appeared morphologically normal and did not exhibit motility defects (*Figure 1D* and *Video 1*).

We found it surprising that deletion of NOCA-1, which has eight isoforms and a critical role in the germline, had such a limited effect on development. Since NOCA-1 has homology to ninein, which has been proposed to anchor microtubules at centrosomes (*Mogensen et al., 2000*; *Delgehyr et al., 2005*), we considered whether it might function redundantly with Patronin, another microtubule minus end-associated protein. To test this, we used a transposon-based method to generate a null mutant in *ptrn-1*, which encodes the only *C. elegans* Patronin family member (*Figure 1C* and *Figure 1—figure supplement 3*; *Frøkjær-Jensen et al., 2010*; *Chuang et al., 2014*). A polyclonal antibody against the PTRN-1 C-terminus recognized a single band of ~130 kD that was absent in *ptrn-1Δ* worms (*Figure 1C*). Like *noca-1Δ* worms, *ptrn-1Δ* worms appeared morphologically normal (*Chuang et al., 2014*; *Marcette et al., 2014*; *Richardson et al., 2014*; *Figure 1D* and *Video 1*). However, in contrast to *noca-1Δ* worms, *ptrn-1Δ* worms were fertile, indicating that PTRN-1 function is not required in the germline.

In striking contrast to the two single mutants, *noca-1Δ; ptrn-1Δ* worms exhibited severe developmental defects. Double mutant worms grew slowly, and ~60% ruptured and died during the first 3 days of post-embryonic development, largely at L4 and early adult stages (*Figure 1D–F* and *Video 1*). The 40% that survived were small and uncoordinated (Small Unc; *Figure 1E*). We conclude that NOCA-1 and PTRN-1 are redundantly required for larval development and viability.

## Patronin and NOCA-1 co-sediment with microtubules from *C. elegans* extracts

Patronin family members bind to microtubule minus ends (*Meng et al., 2008*; *Goodwin and Vale, 2010*; *Hendershott and Vale, 2014*; *Jiang et al., 2014*). To determine if this is also true for *C. elegans*

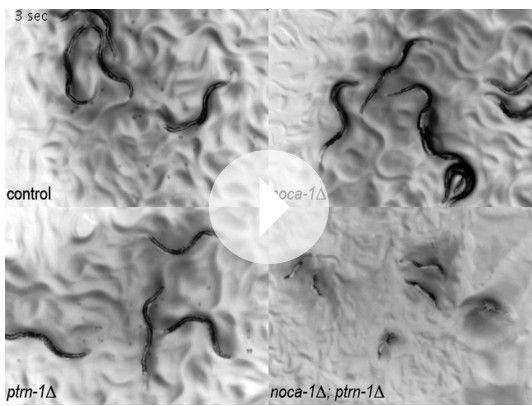

**Video 1.** NOCA-1 and PTRN-1 redundantly perform a function essential for larval development. Worms with the indicated genotypes were filmed using an eyepiece camera (DinoEye) mounted on a dissection scope 72 hr after release from a synchronized L1 stage. Playback is 2× realtime.

PTRN-1, we expressed and purified recombinant GFP fusions with full-length PTRN-1 and DmPatronin, as a control, from insect cells (*Figure 1G*). Employing a kinesin gliding assay to define polarity at physiological ionic strength (100 mM KCl), we observed puncta of GFP::PTRN-1 and GFP::DmPatronin at the leading end of gliding microtubules, indicating binding to minus ends (*Figure 1G*). Thus, *C. elegans* PTRN-1 possesses the minus end recognition activity predicted based on its homology to Patronin family proteins.

Both NOCA-1 and PTRN-1 were detected in the pellet after microtubule sedimentation from *C. elegans* extracts (*Figure 1H*), indicating that NOCA-1 possesses either a direct or indirect microtubule-binding activity. To determine if purified NOCA-1 binds directly to microtubules, we purified GFP-tagged NOCA-1$^{NHD}$ and NOCA-1$^{LICR+NHD}$ from insect cells (*Figure 1—figure supplement 4A*). Hydrodynamic analysis in 500 mM salt indicated that both NOCA-1 fusions were dimeric, whereas GFP-tagged PTRN-1 and DmPatronin were monomeric (*Figure 1—figure supplement 4B–D*). Unfortunately, lowering the ionic strength to physiological levels caused both NOCA-1$^{LICR+NHD}$ and NOCA-1$^{NHD}$ to precipitate. Adding detergents or stabilizers, such as glycerol or sucrose, did not circumvent this problem; however, we were able to generate an MBP::NOCA-1$^{NHD}$::GFP fusion that was soluble at physiological ionic strength. While the ability of this soluble fusion to co-sediment with microtubules was negligible (*Figure 1—figure supplement 4E,F*), we did observe that aggregated forms of NOCA-1 fusion proteins associated with microtubules. When small aggregates of GFP::NOCA-1$^{NHD}$ or GFP::NOCA-1$^{LICR+NHD}$ were analyzed in a coverslip-anchorage assay, analogous to that performed previously for Patronin (*Goodwin and Vale, 2010*; *Figure 1—figure supplement 5A*), they anchored microtubules by binding to their ends (*Figure 1—figure supplement 5A*). Similarly, dilution of MBP::NOCA-1$^{NHD}$::GFP into a classical microtubule assay buffer caused it to form small aggregates that bound along the lengths of microtubules (*Figure 1—figure supplement 5B*). These results hint that NOCA-1 may associate directly with microtubules, although significant additional work will be necessary to overcome the limitations imposed by the low solubility of purified NOCA-1 in order to rigorously assess microtubule interactions in vitro.

## NOCA-1 and Patronin/PTRN-1 control assembly of a circumferential microtubule array required for larval development

The failure of larval development in the *noca-1Δ; ptrn-1Δ* double mutant indicated that NOCA-1 and PTRN-1 function in parallel to promote larval growth and morphogenesis. Mitotic spindle assembly in the early embryo and embryonic viability were not affected by either single or double inhibitions of NOCA-1 and PTRN-1 (*Figure 2—figure supplement 1*), indicating that their redundant function essential for larval development is likely in a differentiated tissue, and not in the formation of centrosomal microtubule arrays required for cell division. To identify this tissue, we expressed PTRN-1::GFP under different tissue-specific promoters. PTRN-1::GFP expressed from its endogenous promoter (P*ptrn-1*) rescued the synthetic lethality of the *noca-1Δ; ptrn-1Δ* double mutant (*Figure 2A*) and localized in multiple tissues, including the larval/adult epidermis, neurons, intestine, and pharynx (*Figure 2—figure supplement 2*). Selective expression of PTRN-1::GFP in the larval/adult epidermis (P*dpy-7*) rescued the lethality and morphology/movement phenotypes of *noca-1Δ; ptrn-1Δ* mutants, whereas no rescue was observed following expression in neurons (P*rgef-1*) or the pharynx and intestine (P*pha-4*) (*Figure 2A* and *Figure 2—figure supplement 2*). Transgenes encoding the NOCA-1d and e isoforms or only the d isoform expressed from their endogenous promoters rescued larval development

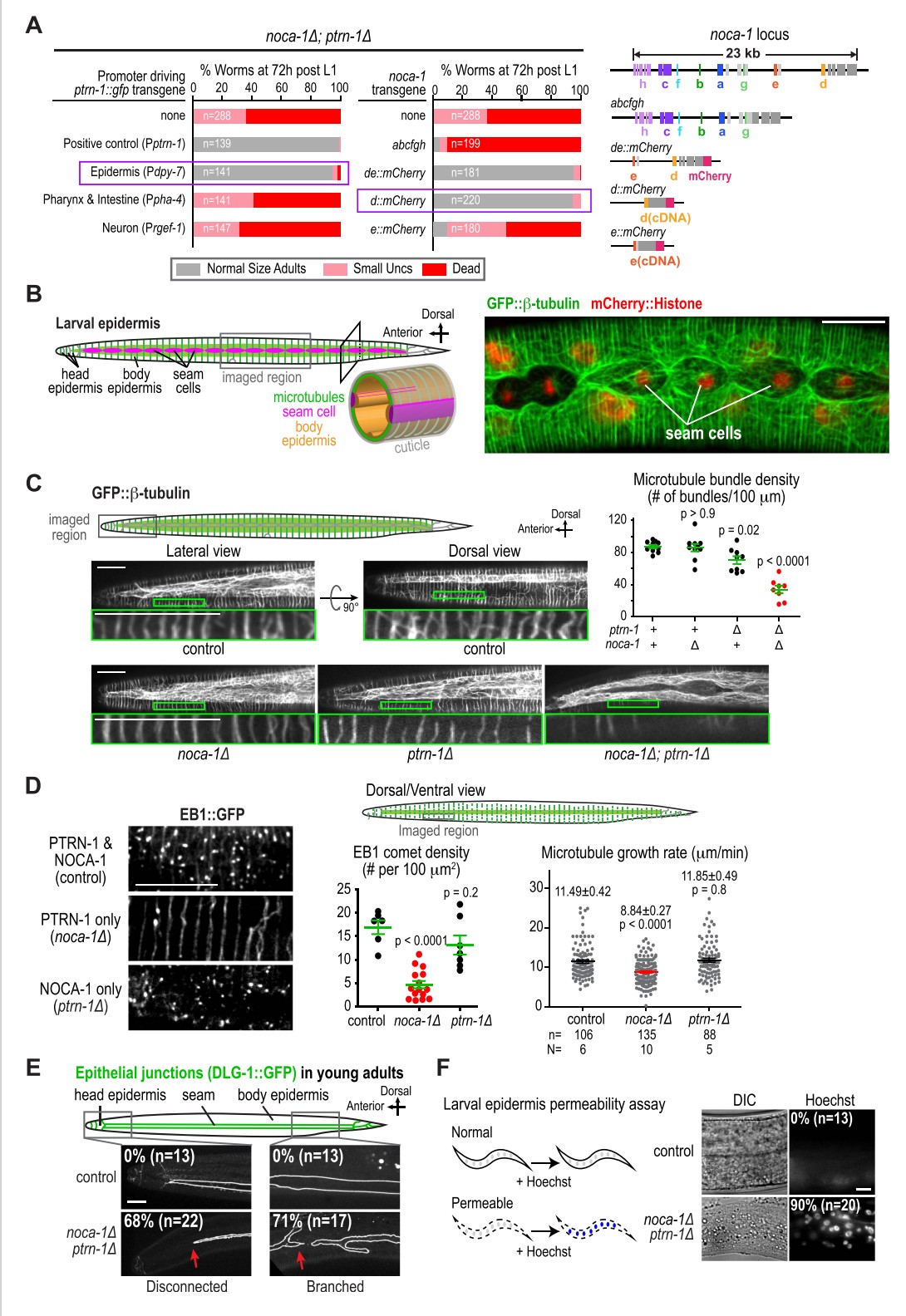

**Figure 2**. NOCA-1 and PTRN-1 control assembly of a circumferential microtubule array required for the integrity of the larval/adult epidermis. (**A**) Left: plots of the percentage of normal-sized adults, small uncs, and dead worms 72 hr post L1 for *noca-1Δ; ptrn-1Δ* worms-expressing PTRN-1::GFP under the control of the indicated promoters or with *noca-1* transgenes directing expression of the indicated isoforms from their own promoters. n is number of worms analyzed in 3–5 independent experiments. Right: schematics of *noca-1* transgenes. Note that the data for *noca-1Δ; ptrn-1Δ* worms in both plots are

*Figure 2. continued on next page*

*Figure 2. Continued*

the same as in *Figure 1E*. (**B**) Left: schematics illustrating the organization of the larval epidermis. The body epidermis (*gold in 3D view*) is a large, thin multinucleated syncytial cell that covers the majority of the worm's body; rows of seam cells (*pink*) are embedded within the body epidermis in rows that run along either side of the worm. Right: maximum intensity projection of fluorescence confocal image of GFP::β-tubulin and mCherry::Histone in the larval epidermis of an L3 stage worm (n = 20). (**C**) Schematic and fluorescence confocal images of L3 stage worms of the indicated genotypes expressing GFP::β-tubulin. Right: plot of microtubule bundle density in worms of the indicated genotypes. (**D**) Left: fluorescence confocal images of L3 stage worms expressing EB1::GFP. Right top: schematic of the imaged region. Right bottom: plots of EB1 comet density and microtubule growth rate in worms of the indicated genotypes. (**E**) Top: schematic of early adult worm expressing DLG-1::GFP, which marks the junctions between the body epidermis and the seam cell syncytia. Bottom: fluorescence confocal images of control and *noca-1Δ; ptrn-1Δ* worms expressing DLG-1::GFP. (**F**) Left: schematic of the permeability assay. Right: DIC and fluorescence images of worms after treatment with Hoechst. Statistics, one-way ANOVA followed by Dunnett's multiple comparisons test. p-values are the probability of obtaining the observed results assuming the test group is the same as control. Error bars are SEM. Scale bars, 10 μm.

The following figure supplements are available for figure 2:

**Figure supplement 1**. Both NOCA-1 and PTRN-1 are dispensable for mitotic divisions.

**Figure supplement 2**. Expression of PTRN-1::GFP in multiple tissues.

**Figure supplement 3**. NOCA-1 immunoblot in lysate from *noca-1Δ* worms expressing *noca-1abcfgh* and the NOCA-1d isoform-specific region is dispensable for its function in the larval epidermis.

**Figure supplement 4**. Illustration of seam cell fusion event at mid-L4 stage.

**Figure supplement 5**. Time course of larval permeability in control and mutant backgrounds.

**Figure supplement 6**. Microtubule bundles in the post-embryonic epidermis co-align with cuticle annuli.

---

in *noca-1Δ; ptrn-1Δ* worms, whereas transgenes encoding the abcfgh (*Figure 2—figure supplement 3*) or e isoforms did not. The short NOCA-1d isoform consists of the NHD and a short unique N-terminal extension (*Figure 1A*). The N-terminal extension was not required for function, since expression of an NHD::GFP fusion under the *Pptrn-1* promoter was sufficient to rescue the double mutant phenotype (*Figure 2—figure supplement 3*). These results indicate that the NHD of NOCA-1 is sufficient to function redundantly with PTRN-1 in the larval/adult epidermis to support organismal growth and development.

The larval/adult epidermis (the worm's 'skin') is composed of a single, multinuclear syncytial cell (hyp7) that covers the majority of the worm's body (gold in 3D schematic in *Figure 2B*). Embedded in this cell are two lateral rows of seams cells that run along either side of the worm's body. The seam cells fuse to form syncytia at the mid-L4 stage (*Chisholm and Hsiao, 2012*; *Figure 2B* and *Figure 2—figure supplement 4*). Other syncytial cells cover the head and tail. We visualized the microtubule array in the syncytial epidermis by co-expressing GFP::β-tubulin and mCherry::histone under control of the *dpy-7* promoter (*Figure 2B*). As previously reported (*Priess and Hirsh, 1986*; *Costa et al., 1997*), the epidermal microtubule array is composed of regularly spaced circumferential bundles that appear as lines perpendicular to the larva/worm body axis in longitudinal sectional views (*Figure 2B*). The density of microtubule bundles along the length of the worm was not significantly different from controls in the *noca-1Δ* mutant and was only slightly reduced in the *ptrn-1Δ* mutant (*Figure 2C* and *Video 2*). In contrast, significantly fewer microtubule bundles were observed in the *noca-1Δ; ptrn-1Δ* double mutant (*Figure 2C* and *Video 2*). We conclude that NOCA-1 and Patronin/ PTRN-1 redundantly control the assembly of a circumferential microtubule array required for larval development.

## NOCA-1 makes the microtubule arrays in the larval/adult epidermis more dynamic

To investigate the impact of NOCA-1 and PTRN-1 on microtubule dynamics, we took advantage of the fact that similarly structured microtubule arrays form in the larval epidermis in the presence of

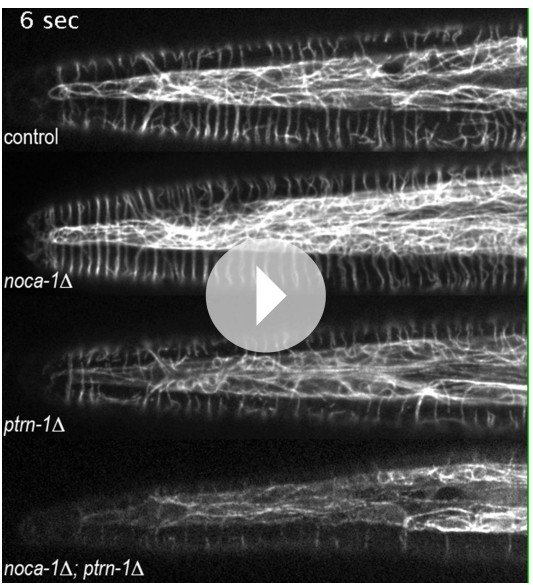

**Video 2.** NOCA-1 and PTRN-1 function in parallel to control microtubule array formation in the larval epidermis. Timelapse fluorescence confocal microscopy was used to acquire images of the head epidermal region of control, *noca-1Δ*, *ptrn-1Δ*, and *noca-1Δ; ptrn-1Δ* worms expressing GFP::β-tubulin. Images were acquired at 1 s intervals. Playback is 6× realtime.

NOCA-1 only, PTRN-1 only, or in the presence of both proteins (*Figure 2C*), and imaged microtubules and growing microtubule ends marked by EB1 comets (*Akhmanova and Steinmetz, 2008*). When only PTRN-1 was present (*noca-1Δ*), microtubules appeared less dynamic than in wild type, whereas microtubules exhibited apparently normal dynamics when only NOCA-1 was present (*ptrn-1Δ*; *Video 2*). Consistent with this impression, the density of EB1 comets was substantially reduced when only PTRN-1 was present (*noca-1Δ*) but was comparable to controls when only NOCA-1 was present (*ptrn-1Δ*; *Figure 2D* and *Video 3*). EB1 signal was observed along the lattice of the bundles and only occasionally in comets when only PTRN-1 was present, possibly due to the reduced number of growing microtubule ends. The microtubule growth rate, measured by tracking of EB1 comets, was also reduced by ~20% compared to controls in worms expressing PTRN-1 only (*noca-1Δ*) but not in worm expressing NOCA-1 only (*ptrn-1Δ*; *Figure 2D*). These results suggest that although either NOCA-1 or PTRN-1 can support the assembly of a circumferential microtubule array in the larval epidermis, the presence of NOCA-1 makes the arrays significantly more dynamic.

## The circumferential microtubule array is required for the integrity of the larval/adult epidermis

To determine if the circumferential microtubule array maintains the structure of the epidermis, we analyzed two features in *noca-1Δ; ptrn-1Δ* double mutants: localization of the apical junction marker DLG-1::GFP (*McMahon et al., 2001*) and integrity of the cuticle, which is secreted by the epidermis to function as an environmental barrier (*Page and Johnstone, 2007*). DLG-1::GFP outlines the junctions between the body epidermis and the seam cell syncytia that are embedded along the left and right sides of the worm

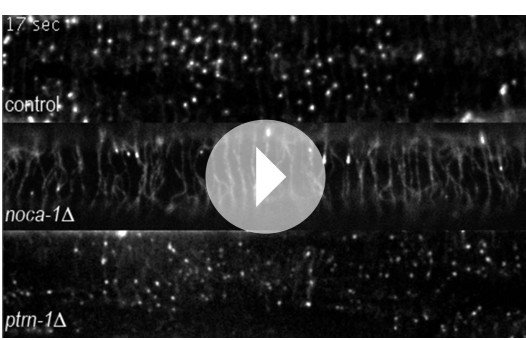

**Video 3.** NOCA-1 makes the microtubule arrays in the larval/adult epidermis more dynamic. Timelapse fluorescence confocal microscopy was used to acquire images of the dorsal or ventral side of larval body epidermis in control, *noca-1Δ*, and *ptrn-1Δ* worms expressing EB1::GFP (marks growing microtubule ends). Images were acquired at 1-s intervals. Playback is 6× realtime.

(*Figure 2—figure supplement 4*). In wild-type worms, parallel lines of DLG-1::GFP are observed running along the entire body length. In contrast, in *noca-1Δ; ptrn-1Δ* double mutants, seam cell syncytia were frequently branched/broken (71%; n = 17) as well as disconnected from the head epidermis (68%; n = 22; *Figure 2E*). This result suggests that the circumferential microtubule array in the body epidermis could have a role in positioning the seam cells prior to fusion. However, since the P*dpy-7* promoter also directs expression in the seam cells, we also cannot rule out that the fusion defect results from direct effects on the seam cells or their capacity to fuse. In addition to seam cell defects, the cuticles of *noca-1Δ; ptrn-1Δ* mutant worms became permeable to the normally excluded Hoechst dye, beginning ~24 hr after the L1 larval stage (*Figure 2F* and *Figure 2—figure supplement 5*). These defects in the epidermis and cuticle likely underlie the rupture phenotype

with extrusion of internal tissues observed in *noca-1Δ; ptrn-1Δ* mutant worms (*Figure 1D*). We conclude that NOCA-1 and PTRN-1 function in parallel to promote the assembly of a circumferential array of microtubule bundles that is required for the morphology and integrity of the larval/adult epidermis.

## γ-tubulin functions together with NOCA-1 and in parallel to Patronin/PTRN-1 to promote larval development and viability

Our analysis placed NOCA-1 and PTRN-1 in parallel pathways controlling assembly of an essential circumferential microtubule array in the embryonic epidermis. Imaging of a GFP fusion with NOCA-1 in the larval epidermis revealed that it had a localization pattern very similar to that of γ-tubulin; NOCA-1 and γ-tubulin were both observed in puncta in the epidermal syncytium where the microtubule bundles are present (*Figure 3A*, magnified insets) and also concentrated along the junctions between the epidermal body syncytium and the seam cells (*Figure 3A,B*). The localization pattern of PTRN-1::GFP was distinct. Consistent with prior work (*Jiang et al., 2014*) PTRN-1::GFP was observed in stretches as well as puncta in the body syncytium. PTRN-1::GFP was also observed in puncta within the seam cells but did not accumulate along the seam cell junctions. In double label images of NOCA-1d::GFP or PTRN-1::GFP with tagRFP::β-tubulin, many puncta of both proteins were observed coincident with the microtubule bundles in the body epidermis (*Figure 3—figure supplement 1*).

Given their similar localization patterns and the fact that knockdown of NOCA-1 and γ-tubulin resulted in an essentially identical defect in the germline (*Green et al., 2011* and *Figure 4* below), we wanted to test whether γ-tubulin functioned in microtubule generation pathways with NOCA-1, PTRN-1, or both in the larval epidermis. Since γ-tubulin is essential for cell division, analyzing its role in the larval epidermis required eliminating γ-tubulin function after the tissue is already formed. To achieve this, we developed a method based on two previously described protein degradation methods (*Caussinus et al., 2012*; *Armenti et al., 2014*) for tissue-specific degradation of a functional GFP-fused target protein. Since fluorescently tagged γ-tubulin fusions were not fully functional (*not shown*), we used a CRISPR/Cas9-mediated strategy (*Dickinson et al., 2013*) to insert a C-terminal GFP tag in the endogenous locus of *gip-2* (*Figure 3—figure supplement 2*), which encodes an essential component of the *C. elegans* γ-tubulin complex (*Hannak et al., 2002*). Endogenously tagged GIP-2 fully supported the essential functions of the γ-tubulin complex, as indicated by the normal development of worms homozygous for the insertion. To specifically degrade GIP-2::GFP in the epidermis, we expressed a GFP nanobody::ZIF-1 fusion under an epidermal promoter (P*dpy-7*; *Figure 3—figure supplement 2*). This fusion, which we call *epiDEG*, serves as a GFP-to-ligase adapter that recognizes GFP-fused target proteins and brings them to the ECS (Elongin-C, Cul2, SOCS-box family) E3 ubiquitin ligase complex for ubiquitination and proteasome-mediated degradation (*Figure 3C*; *DeRenzo et al., 2003*). Quantification revealed that the GIP-2::GFP signal in the larval epidermis was reduced by >80% compared to controls in *epiDEG* worms whereas the signal in the germline was unaffected (*Figure 3D*), indicating the GFP-mediated degradation is efficient and tissue specific. The *gip-2::gfp; epiDEG* animals grew slightly slower than wild-type worms and a small percentage of them were arrested at early larval stage (*Figure 3E–G* and *Video 4*), possibly because the *dpy-7* promoter-driven *epiDEG* may cause some degradation of GIP-2::GFP in the dividing seam cells. However, the majority of *gip-2::gfp; epiDEG* animals exhibited normal development.

Having established a method to selectively degrade an essential γ-tubulin complex subunit in the larval epidermis, we tested whether this perturbation of γ-tubulin enhanced the *noca-1Δ* or *ptrn-1Δ* phenotypes. We found that *noca-1Δ; gip-2::gfp; epiDEG* animals exhibited the same mild phenotypes observed in *gip-2::gfp; epiDEG* animals. In contrast, more than 70% of *ptrn-1Δ; gip-2::gfp; epiDEG*; animals ruptured and died at late L4 to early adult stages (*Figure 3E–G* and *Video 4*). The 30% survivors were mostly small and uncoordinated or arrested as larva. This striking difference between the effects of inhibiting the γ-tubulin complex in the *noca-1Δ* and *ptrn-1Δ* mutants suggests that the γ-tubulin complex functions together with NOCA-1 and in parallel to PTRN-1 for non-centrosomal microtubule array generation in the larval epidermis (*Figure 3H*).

## NOCA-1 and γ-tubulin, but not PTRN-1, are required for the function of non-centrosomal microtubule arrays that position nuclei in the germline

A major phenotypic difference between *noca-1Δ* and *ptrn-1Δ* worms is that the former are sterile whereas the latter are fertile (*Figure 4B*). The *C. elegans* germline is a syncytial structure in which

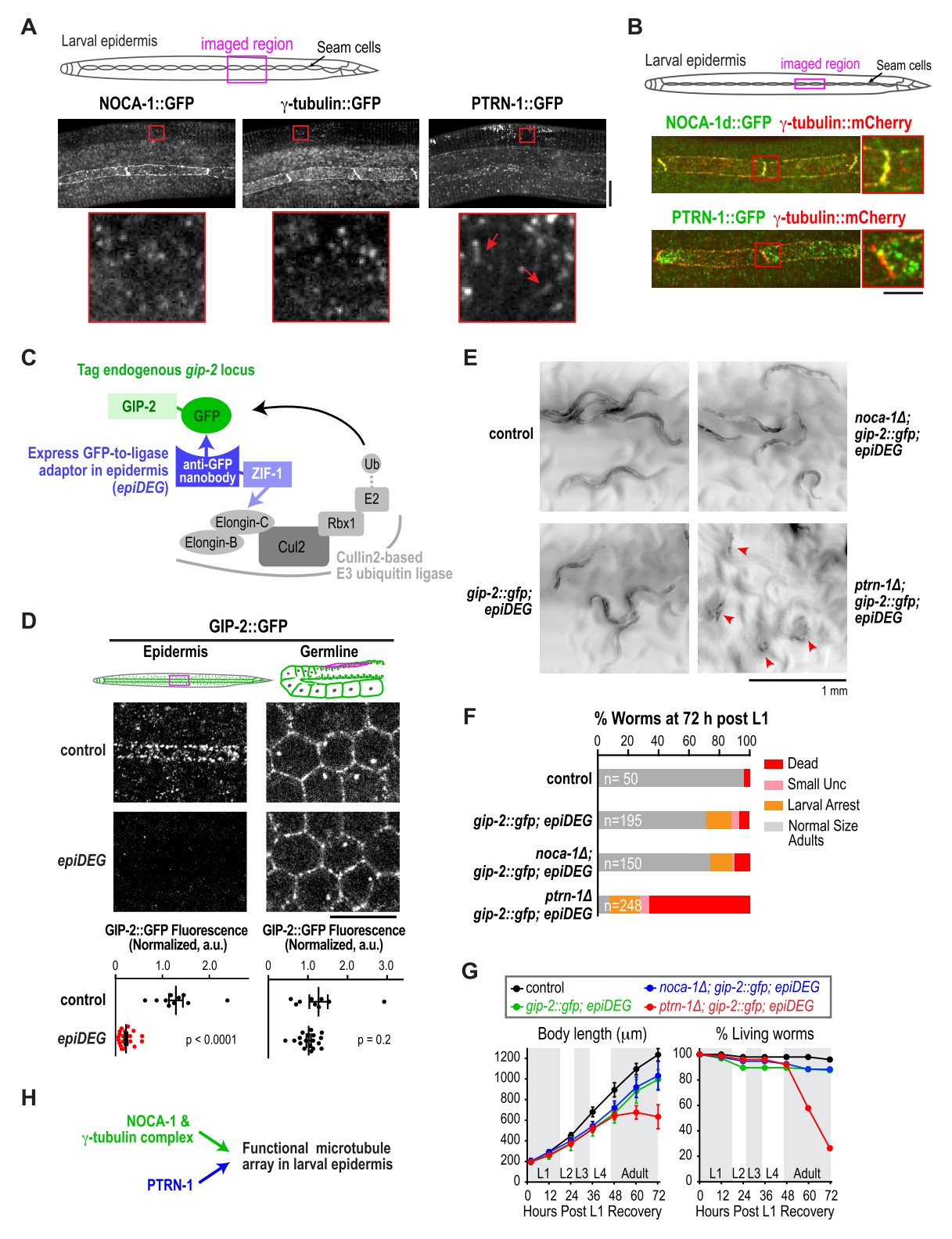

**Figure 3**. The γ-tubulin complex functions coordinately with NOCA-1 and in parallel to PTRN-1 to promote larval development and viability. (**A**) Top: schematic of the imaged region. Bottom: fluorescence confocal images of L3 stage worms expressing NOCA-1::GFP (n = 27), γ-tubulin::GFP (n = 6), or PTRN-1::GFP (n = 17). Insets below are magnified eightfold. Arrowheads point to examples of stretches observed in worms expressing PTRN-1::GFP. Note that the vertical lines in the images are cuticle auto-fluorescence due to high laser power and long exposure times required to visualize the GFP

*Figure 3. continued on next page*

Figure 3. Continued

puncta/stretches. (**B**) Top: schematic of the imaged region. Bottom: fluorescence confocal images of L3-stage worms co-expressing NOCA-1d::GFP (n = 12) or PTRN-1::GFP (n = 4) with γ-tubulin::mCherry. (**C**) Schematic outlining the method used to specifically degrade the essential γ-tubulin complex component GIP-2::GFP in the epidermis. (**D**) Top: schematics and fluorescence confocal images of L4 stage worms expressing GIP-2::GFP with or without Pdpy-7::GFP nanobody::ZIF-1 (epiDEG). Bottom: plots of normalized GIP-2::GFP fluorescence intensity in the epidermis or germline from worms with indicated genotypes. (**E**) Images of control and mutant worms 72 hr post L1 recovery (snapshots from *Video 4*). Arrowheads mark dead worms. (**F**) Plot of percentage of normal-sized adults, larval arrest, small uncs, and dead worms 72 hr post L1 for the indicated genotypes. n is total number of worms analyzed in 1 (control), 2 (*gip-2::gfp; epiDEG* and *gip-2::gfp; epiDEG ;noca-1Δ*), or 3 (*gip-2::gfp; epiDEG; ptrn-1Δ*) independent experiments. (**G**) Plots of body length (*left*) and % living worms (*right*) vs time for worms with the indicated genotypes. (**H**) Schematic describing two parallel pathways for assembly of a functional microtubule array in larval epidermis. Statistics, Student's *t*-test. p-values are the probability of obtaining the observed results assuming the test group is the same as control. Error bars are SEM. Scale bars, 10 μm or as indicated.

The following figure supplements are available for figure 3:

**Figure supplement 1**. NOCA-1 and PTRN-1 localize along microtubules in the larval epidermis.

**Figure supplement 2**. Strategy to selectively inhibit the γ-tubulin complex in the larval/adult epidermis of *C. elegans*.

nuclei in various stages of meiotic prophase are housed in membrane-bound compartments that are open on one side towards the common cytoplasmic core, called the rachis. Non-centrosomal microtubule arrays assemble within the compartments that hold the nuclei near the surface and prevent them from dropping into the rachis (*Figure 4A*; *Zhou et al., 2009*). Within the rachis there are also microtubules that flow with the streaming cytoplasm into the forming oocytes (*Wolke et al., 2007*). Imaging germline architecture in worms expressing a GFP-tagged plasma membrane probe along with mCherry-histone or GFP::β-tubulin revealed that *noca-1* deletion resulted in an essentially identical phenotype to γ-tubulin depletion; in both cases, nuclei fell out of their compartments and formed clumps in the rachis center, indicating a dramatic failure in the function of the microtubule arrays in the compartments (*Figure 4B,C*). In contrast, germline structure in *ptrn-1Δ* worms was similar to that in controls (*Figure 4B,C*). Since compartment structure collapsed as the nuclei fell into the rachis, we could not assess the impact of loss of NOCA-1 or γ-tubulin on the dynamics of the arrays within the compartments. However, we were able to measure the density of growing microtubule plus ends, measured as the number of EB1 comets, in a fixed area of the rachis, which was reduced to a similar extent by NOCA-1 and γ-tubulin inhibitions (*Figure 4D* and *Video 5*).

Consistent with lack of an effect of *ptrn-1Δ*, PTRN-1 is not expressed in the germline (*not shown*); in addition, forcing PTRN-1 expression in the germline did not rescue *noca-1Δ* sterility (*Figure 4—figure supplement 1*). Selectively depleting the longest NOCA-1 isoform (NOCA-1h) using a dsRNA targeting its unique N-terminal extension disrupted germline architecture and led to sterility (*Figure 4E*), and expressing NOCA-1h from an RNAi-resistant transgene under its own promoter rescued both phenotypes (*Figure 4F* and *Figure 4—figure supplement 2*), indicating that NOCA-1h is both necessary and sufficient for germline function. Expression of a NOCA-1 truncation that included the NHD and the long isoform common region (NOCA-1$^{NHD+LICR}$) under the same promoter also rescued the effects of depleting NOCA-1h on the germline, whereas expression of the NHD alone did not (NOCA-1$^{NHD}$; *Figure 4G*). Thus, in the germline, NOCA-1 function requires the LICR in addition to the NHD, but the h isoform specific region is not essential. We conclude that, in the germline, γ-tubulin and NOCA-1h act independently of PTRN-1 to direct assembly of non-centrosomal microtubule arrays that position nuclei.

## γ-tubulin contributes to the cell surface recruitment of NOCA-1 in the germline

In the germline, NOCA-1h co-localizes with γ-tubulin to the surface of the compartments but does not co-localize with γ-tubulin at centrosomes (*Figure 5A,B*, red arrows point to centrosomes). This result raised the possibility that NOCA-1 promotes non-centrosomal microtubule array formation by recruiting γ-tubulin to the cell surface. We tested this possibility by imaging γ-tubulin::mCherry in *noca-1Δ* worms. Although compartment structure is disrupted in *noca-1Δ* worms, γ-tubulin::mCherry

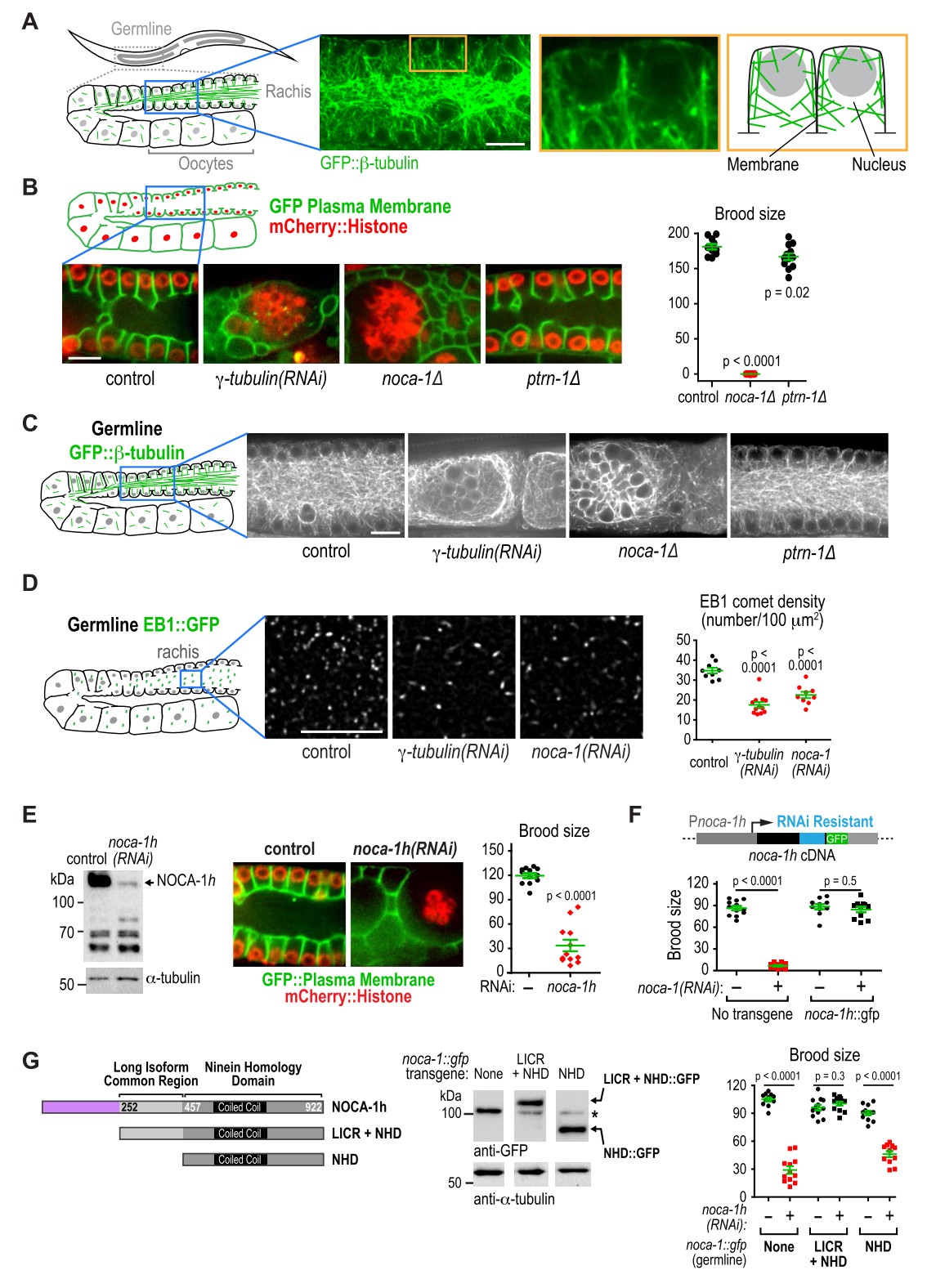

**Figure 4**. NOCA-1 isoform h functions in the germline to assemble a non-centrosomal microtubule array for nuclear positioning. (**A**) Left: schematic showing the germline and location of the imaged region. Middle: fluorescence confocal image of the germline in a worm expressing GFP::β-tubulin. Inset to the right is magnified 3.3-fold. Right: Schematic of the region highlighted in the inset, illustrating the organization of the microtubule arrays in the compartments that hold the nuclei near the cell surface and prevent them from falling into the rachis. (**B**) Left top: schematic illustrating the structure of the

*Figure 4. continued on next page*

*Figure 4. Continued*

syncytial germline. Left bottom: fluorescence confocal images of germlines in control (n = 14), γ-tubulin(RNAi) (n = 7), noca-1Δ (n = 12), and ptrn-1Δ (n = 11) worms expressing a GFP-tagged plasma membrane marker and mCherry-tagged histone H2B. Frequencies of disorganized germlines with nuclei falling out of their compartments were 100% in γ-tubulin(RNAi) and noca-1Δ worms and 0% in control and ptrn-1Δ worms. Right: plot of brood size for worms of the indicated genotypes. (**C**) Left: schematic illustrating microtubule organization in the germline. Right: fluorescence confocal images of germlines in control (n = 22), γ-tubulin (RNAi) (n = 10), noca-1Δ (n = 13) and ptrn-1Δ (n = 7) worms expressing GFP::β-tubulin. Frequencies of the nuclear fall-out phenotype were 100% in γ-tubulin(RNAi) and noca-1Δ worms and 0% in control and ptrn-1Δ worms. (**D**) Left: schematic showing the location of the imaged region. Middle: fluorescence confocal images of growing microtubule ends marked by EB1::GFP in the germline. Right: plot of EB1 comet density in worms depleted of the indicated proteins by RNAi. (**E**) Left: immunoblot of NOCA-1 in lysates from control and noca-1h(RNAi) worms. Middle: fluorescence confocal images of germlines in control (n = 13) and noca-1h (RNAi) (n = 10) worms expressing a GFP-tagged plasma membrane marker and mCherry::histone. Frequencies of disorganized germlines with nuclear fallout were 100% in noca-1h(RNAi) and 0% in control worms. Right: plot of brood size for control and noca-1h(RNAi) worms. (**F**) Top: schematic illustrating the RNAi-resistant noca-1h::gfp transgene. Bottom: brood size plot for worms subjected to the indicated perturbations. (**G**) Left: schematic showing NOCA-1h and the two analyzed truncations. Germline expression was driven by the noca-1h promoter. Middle: immunoblot of lysates prepared from worms with the indicated genotypes. The asterisk marks a non-specific band. Right: Plot of brood size for worms subjected to indicated perturbations. Statistics in **B** and **D**, one-way ANOVA followed by Dunnett's multiple comparisons test. Statistics in **E**, **F** and **G**, Student's *t*-test. p-values are the probability of obtaining the observed results assuming the test group is the same as control. Error bars are SEM. Scale bar, 10 μm.

The following figure supplements are available for figure 4:

**Figure supplement 1**. Ectopic germline expression of PTRN-1 does not substitute the germline function of NOCA-1.

**Figure supplement 2**. Sequence of the RNAi-resistant region of the NOCA-1h::GFP transgene and localization of NOCA-1h::GFP in the germline.

was still clearly observed on the compartment surfaces indicating that NOCA-1 is not required to recruit γ-tubulin to that location (*Figure 5C*).

We next tested if NOCA-1 required γ-tubulin to localize to the surface of germline compartments. As full-length NOCA-1h and NOCA-1[LICR+NHD], which lack the h isoform-specific region, are both functional, we analyzed the localization of both in control and γ-tubulin-depleted germlines. Surprisingly, NOCA-1[LICR+NHD] required γ-tubulin to localize to compartment surfaces whereas full-length NOCA-1h did not (*Figure 5D,E*). This result suggested that the non-essential isoform-specific region of NOCA-1h harbors a γ-tubulin-independent cell surface targeting activity (*Figure 5H*). Consistent with this idea, a GFP fusion with the h isoform specific region localized to compartment surfaces, and this localization was dependent on a predicted palmitoylation site (cysteine 10; *Figure 5—figure supplement 1*). Mutation of this predicted palmitoylation site in the full-length protein (NOCA-1h[C10A]) did not compromise NOCA-1h function but rendered its localization γ-tubulin dependent (*Figure 5F–H*, *Figure 5—figure supplement 1*). This result explains why NOCA-1h localization at compartment surfaces was not eliminated by γ-tubulin depletion and implicates a potential lipid modification in providing a redundant means for NOCA-1 targeting to the membrane.

γ-tubulin could direct NOCA-1 localization to the cell surface either through a direct interaction or indirectly through nucleated microtubules. To distinguish these two possibilities, we disrupted microtubule assembly by using RNAi to deplete α-tubulin. While this disrupted germline structure to a comparable extent to inhibition of NOCA-1 or γ-tubulin, cell surface targeting of NOCA-1h[C10A] was still observed (*Figure 5F*). This result suggests that an interaction between NOCA-1[NHD+LICR] and the γ-tubulin complex may contribute to recruitment of NOCA-1 to the cell surface. However, we have not yet detected an interaction in immunoprecipitations from *C. elegans* extracts or yeast two-hybrid experiments with NOCA-1 and γ-tubulin complex components, indicating that additional work is needed to understand precisely how γ-tubulin promotes the cell surface recruitment of NOCA-1. Based on these results, we conclude that the γ-tubulin complex recruits NOCA-1 to the cell surface, where they are both required to generate functional non-centrosomal microtubule arrays that position nuclei within compartments.

## NOCA-1, but not PTRN-1, is required for the function of a non-centrosomal microtubule array that positions nuclei in the embryonic epidermis

Our prior work suggested that NOCA-1 is also involved in assembly of non-centrosomal microtubule arrays that position nuclei in the embryonic epidermis (*Green et al., 2011*). Imaging of noca-1Δ

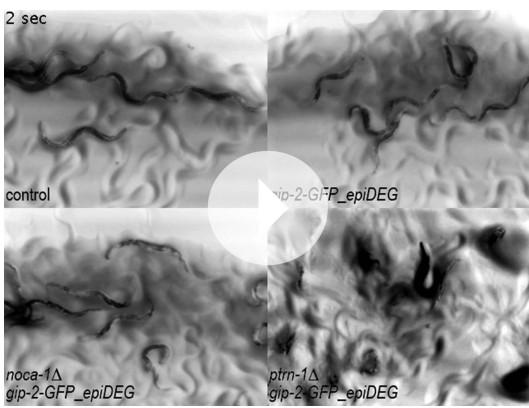

**Video 4.** Epidermal degradation of GIP-2::GFP synergizes with *ptrn-1Δ* but not *noca-1Δ*. Worms with the indicated genotypes were filmed using an eyepiece camera (DinoEye) mounted on a dissection scope 72 hr after release from a synchronized L1 stage. Playback is 2× realtime.

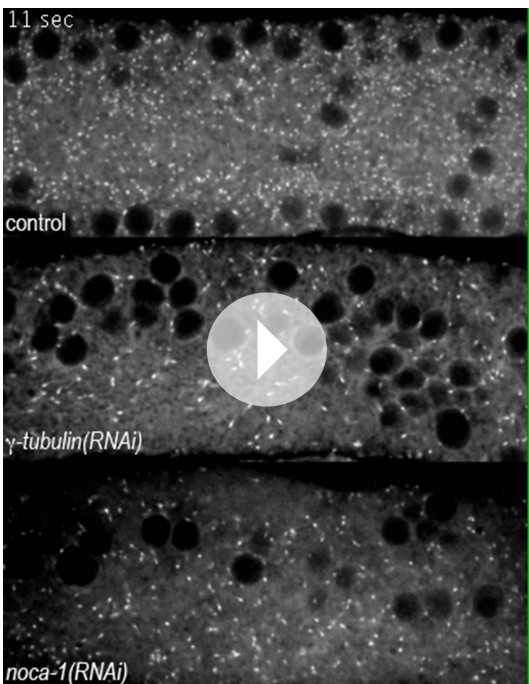

**Video 5.** Depletion of γ-tubulin or NOCA-1 reduces growing microtubule ends in the germline. Timelapse fluorescence confocal microscopy was used to acquire images of a central plane of the pachytene region of the germline in worms expressing EB1::GFP (marks growing microtubule ends). Images of control, *γ-tubulin(RNAi)*, and *noca-1(RNAi)* worms were acquired at 1-s intervals. Playback is 6× realtime.

mutant embryos expressing GFP::β-tubulin suggested a reduction in the number of microtubules in the embryonic epidermis (*Figure 6A*). Consistent with this, EB1 comet density was also reduced ~twofold in *noca-1Δ* embryos (*Figure 6B* and *Video 6*). The microtubule arrays in these cells have previously been implicated in nuclear migration (*Fridolfsson and Starr, 2010*; *Starr and Fridolfsson, 2010*). Defects in nuclear migration lead to the presence of nuclei in the larval dorsal cord, which is not observed in wild type (*Figure 6C*; *Fridolfsson and Starr, 2010*). A clear nuclear migration defect was observed in *noca-1Δ* mutants, whereas no defect was observed in *ptrn-1Δ* mutants (*Figure 6C*). NOCA-1 co-localizes with γ-tubulin in the embryonic epidermis (*Figure 6D*). Although we do not yet have a tissue-specific knockdown system to determine if γ-tubulin is required for assembly of this array, these results suggest that NOCA-1 functions with γ-tubulin independently of PTRN-1 in the embryonic epidermis as it does in the germline.

In the embryonic epidermis, a *noca-1* transgene encoding the *abcfgh* isoforms (*Figure 2—figure supplement 3A*) rescued nuclear migration, whereas a comparable transgene with a stop codon that specifically blocks expression of the *b* isoform (*a\*cfgh*) did not (*Figure 6E*). Expression of the *b* isoform under an epidermal promoter rescued nuclear migration, identifying NOCA-1b as necessary and sufficient for NOCA-1 function in the embryonic epidermis (*Figure 6E*). A truncation analysis revealed that although expression of NOCA-1[NHD] appeared to partially suppress the nuclear migration defect, expression of NOCA-1[NHD+LICR] was required for full rescue. We conclude that, as in the germline, a long NOCA-1 isoform that includes the LICR as well as the NHD is required to direct the PTRN-1-independent assembly of a functional non-centrosomal microtubule array that positions nuclei in the embryonic epidermis.

## Discussion

The remarkable diversity of microtubule arrays in differentiated tissues has been appreciated for quite some time (*Keating and Borisy, 1999*; *Bartolini and Gundersen, 2006*). However, as the major focus has been on centrosomal arrays that are present in dividing cells, the molecular mechanisms underlying this diversity are just beginning to be explored. Here, our analysis in *C. elegans* has revealed an essential role for the ninein-related protein NOCA-1 in the formation of functional

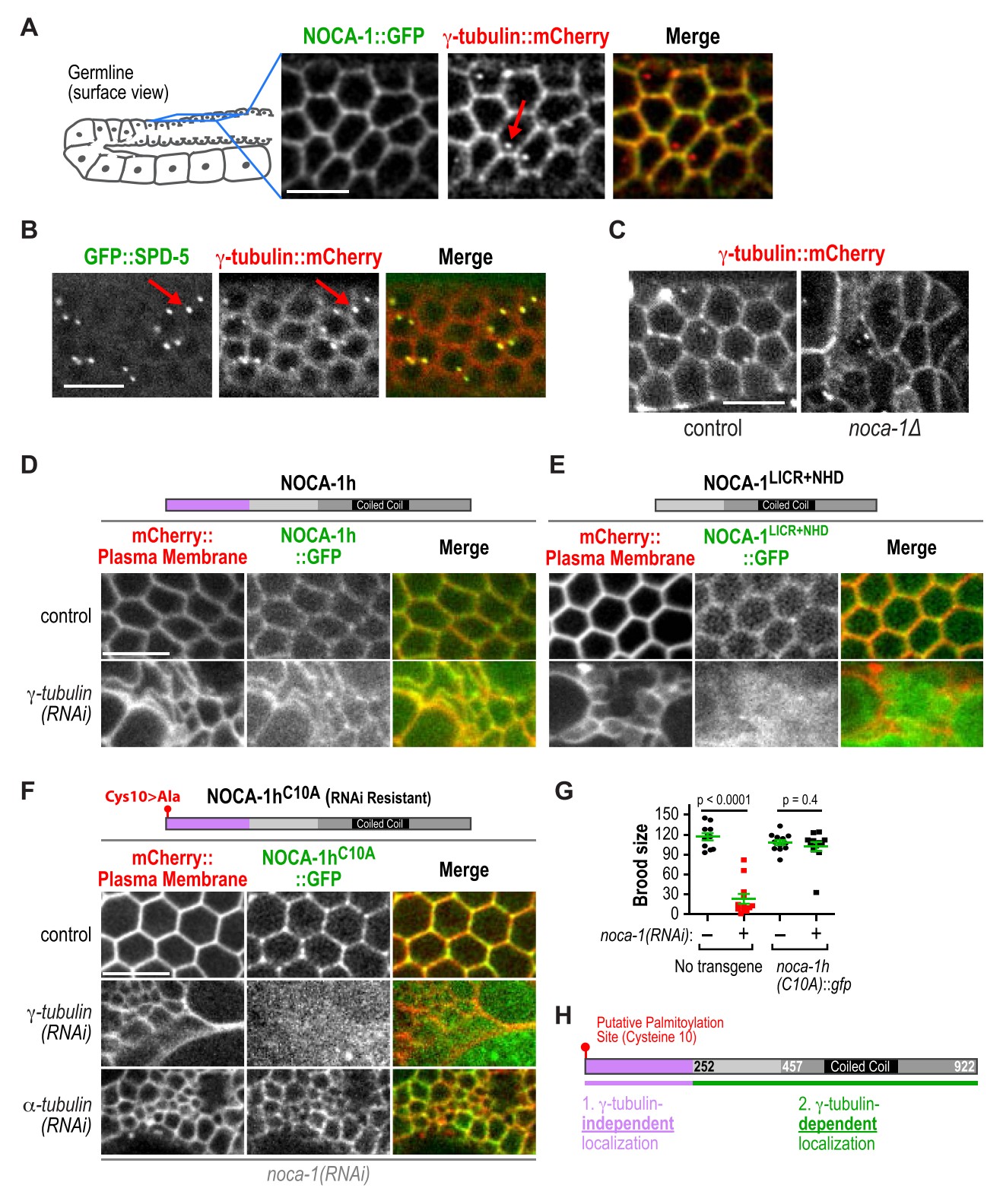

**Figure 5**. γ-tubulin-dependent and independent mechanisms target NOCA-1 to the plasma membrane in the germline. (**A**) Left: schematic of region imaged in **A**–**F**. Right: fluorescence confocal images of the germline in worms co-expressing NOCA-1::GFP and γ-tubulin::mCherry (n = 10). Arrow points to a centrosome. (**B**) Fluorescence confocal images of a germline in a worm co-expressing GFP::SPD-5 (a centrosome marker) and γ-tubulin::mCherry (n = 13). Arrows point to centrosomes. (**C**) Fluorescence confocal images of γ-tubulin::mCherry in the germline of control (n = 11) and *noca-1Δ* (n = 8) worms.
*Figure 5. continued on next page*

*Figure 5. Continued*

(**D**) Fluorescence confocal images of the germline in control (n = 16) and γ-tubulin(RNAi) (n = 10) worms co-expressing NOCA-1h::GFP and an mCherry-tagged plasma membrane marker. (**E**) Fluorescence confocal images of the germline from control (n = 25) and γ-tubulin(RNAi) (n = 23) worms expressing NOCA-1h[LICR+NHD]::GFP and an mCherry-tagged plasma membrane marker. (**F**) Fluorescence confocal images of the germline in worms expressing NOCA-1h[C10A]::GFP and an mCherry-tagged plasma membrane marker that were depleted of endogenous NOCA-1 by RNAi. Images are shown for control worms (n = 17) or worms that were also depleted of γ-tubulin (n = 20) or α-tubulin (n = 18). (**G**) Top: schematic illustrating the RNAi-resistant NOCA-1h[C10A]::GFP transgene. Bottom: brood size plot for worms subjected to indicated perturbations. (**H**) Schematic summarizing the mechanisms that target NOCA-1h to the cell surface in the germline. Statistics, Student's *t*-test. p-values are the probability of obtaining the observed results assuming the test group is the same as control. Scale bars, 10 μm.

The following figure supplement is available for figure 5:

**Figure supplement 1**. The isoform specific region of NOCA-1h localizes to the plasma membrane through a putative palmitoylation.

non-centrosomal arrays in three different differentiated tissues (*Figure 7A*). Direct phenotypic comparisons and controlled ablation following differentiation revealed a close collaboration between NOCA-1 and γ-tubulin in non-centrosomal array formation (*Figure 7A*). In one of the three tissues we examined (the larval epidermis), NOCA-1/γ-tubulin acted in parallel to the microtubule minus end-binding factor PTRN-1 (*Figure 7A*). Based on our results, we propose that the formation of functional non-centrosomal arrays involves coordination of ninein family proteins and γ-tubulin, acting in a parallel and potentially mechanistically distinct manner from the Patronin family of microtubule minus end-binding proteins.

## NOCA-1 relationship to vertebrate ninein

A common region of all 8 NOCA-1 isoforms shares homology with a region of vertebrate ninein that has been implicated in microtubule anchoring at centrosomes (*Delgehyr et al., 2005*); this region is absent in the homologous ninein-like protein that is also present in vertebrates. Like NOCA-1, ninein has been shown to re-localize to the cell surface during the assembly of non-centrosomal microtubule arrays in simple and stratified epithelia (*Mogensen et al., 2000*; *Lechler and Fuchs, 2007*; *Moss et al., 2007*), suggesting a role in the assembly of non-centrosomal microtubule arrays. Our results show that in the germline and embryonic epidermis NOCA-1 and γ-tubulin are required independently of Patronin/PTRN-1, whereas in the larval/adult epidermis, the NOCA-1/γ-tubulin pathway and the Patronin-dependent pathway redundantly support microtubule generation (*Figure 7A*). We expect that our analysis of NOCA-1 may inform studies of vertebrate ninein, mutations in which have been implicated in the human disorders microcephalic primordial dwarfism and spondyloepimetaphyseal dysplasia (*Dauber et al., 2012*; *Grosch et al., 2013*). The functional overlap between the ninein and Patronin families of microtubule cytoskeleton-associated proteins observed in the larval/adult epidermis may also aid future analysis of these two protein classes in vertebrates.

## Models for the coordinated action of NOCA-1 and γ-tubulin

Our data suggest that NOCA-1 functions together with γ-tubulin to promote the formation of non-centrosomal microtubule arrays in multiple tissues. We identified three NOCA-1 isoforms that are each necessary and sufficient to promote the assembly of different non-centrosomal microtubule arrays (*Figure 7A*). This pattern suggests that the remaining five NOCA-1 isoforms will function with γ-tubulin in the assembly of microtubule arrays in tissues that we have not yet characterized; some of these may also act in parallel to PTRN-1. Importantly, the isoform-specific regions were not essential for NOCA-1 function in the three different contexts analyzed, suggesting that these regions primarily reflect use of alterative promoters/transcriptional start sites. In the germline, the tissue-specific isoform region directed non-essential, γ-tubulin-independent membrane localization, potentially via palmitoylation of a cysteine residue in the extreme N-terminus. Whether this residue is indeed palmitoylated will need to be addressed in future work.

In the tissues we analyzed, NOCA-1 co-localized with γ-tubulin (except at centrosomes) and NOCA-1 inhibition phenocopied inhibition of γ-tubulin, blocking the key functions of the arrays and

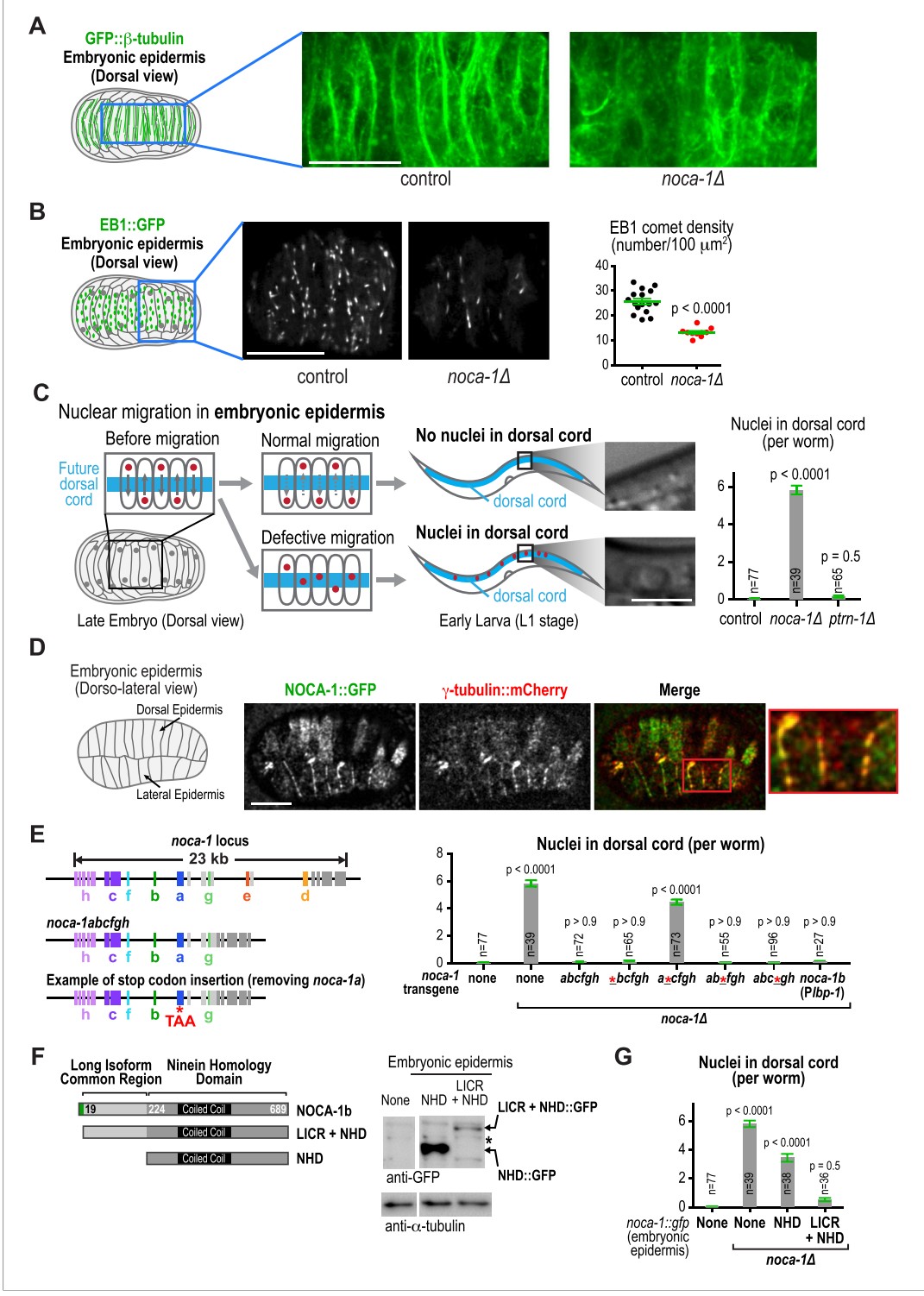

**Figure 6**. NOCA-1, but not PTRN-1, is required for the function of a non-centrosomal microtubule array that positions nuclei in the embryonic epidermis. (**A**) Left: schematic showing the imaged region of the dorsal embryonic epidermis. Right: maximum intensity projections of fluorescence confocal images of the dorsal epidermis in control (n = 4) and *noca-1Δ* (n = 5) embryos expressing GFP::β-tubulin. Images were captured and displayed using identical settings. (**B**) Left and Middle: schematic and images of control (n = 16) and *noca-1Δ* (n = 10) embryos expressing EB1::GFP to mark growing microtubule ends. Right: plot of EB1 comet density in control and *noca-1Δ* embryos. (**C**) Left: schematic illustrating nuclear migration in the developing dorsal epidermis of *C. elegans* embryos. Right:

*Figure 6. continued on next page*

*Figure 6. Continued*

plot of the number of nuclei in the dorsal cord for worms with indicated genotypes. (**D**) Left: schematic showing location of the imaged region. Right: images of *C. elegans* embryos co-expressing NOCA-1::GFP and γ-tubulin:: mCherry (n = 14). (**E**) Left: schematic illustrating *noca-1* transgenes expressing different isoform subsets. 2.4 kb of 5′ UTR and 1.2 kb of 3′ UTR were used in all transgenes. Right: plot of nuclei number in dorsal cord for worms with indicated genotypes. P*lbp-1* is an epidermis specific promoter. Data for control and *noca-1Δ* are the same as in (**C**). (**F**) Left: schematic of the two analyzed truncations. Embryonic epidermis expression was driven by P*lbp-1*. Right: GFP immunoblot of worm lysates prepared from worms with indicated genotypes. '*' marks a non-specific band. (**G**) Plot of nuclei number in dorsal cord for worms with indicated genotypes. Note that data for control and *noca-1Δ* are the same as in (**C**) and (**E**). Error bars are SEM. Statistics in **C**, **E** and **G**, one-way ANOVA followed by Dunnett's multiple comparisons test. Statistics in **B**, Student's *t*-test. p-values indicate the probability of obtaining the observed results assuming the test group is the same as control. Scale bars, 10 μm.

leading to a similar reduction in the number of EB1-marked growing microtubule ends. In the germline, where we were able to analyze localization dependencies, γ-tubulin localized to the cell surface independently of NOCA-1. Understanding how γ-tubulin is recruited to non-centrosomal sites is an important question, as SPD-5, the major pericentriolar material matrix component that is thought to recruit γ-tubulin to centrosomes, is not recruited to non-centrosomal sites (*Figure 5B*; *Feldman and Priess, 2012*).

In contrast to the NOCA-1-independent targeting of γ-tubulin complexes, a functional version of NOCA-1 lacking the putative palmitoylation site, required γ-tubulin for its cell surface targeting. Depleting α-tubulin, while having a comparable effect to γ-tubulin removal on germline structure, did not disrupt NOCA-1 targeting. This result suggests that NOCA-1 may be recruited to the surface via an interaction with γ-tubulin rather than the microtubules that it nucleates, although we cannot fully exclude a contribution from residual microtubules in the α-tubulin depletion.

Our functional analysis raises the important mechanistic question of how γ-tubulin and NOCA-1 act together. One model is that γ-tubulin complexes at non-centrosomal sites recruit NOCA-1, which in turn activates their nucleating activity, leading to generation of new microtubules. Structural work on γ-tubulin containing complexes has suggested that their activation may be coupled to interaction with factors that recruit them to specific sites (*Kollman et al., 2011*). Since *C. elegans*, like budding yeast, has components of the γTuSC (γ-tubulin small complex) but not the γTuRC (γ-tubulin ring complex), one possibility is that NOCA-1 would drive assembly of the γTuSC into larger γTuRC-like complexes as proposed for γTuSC-anchoring factors in budding yeast (*Figure 7B*; *Kollman et al., 2011*). A second model is that NOCA-1 is recruited by γ-tubulin to generate a structure that stabilizes and/or anchors nascent microtubule minus ends generated by γ-tubulin's nucleating activity (*Figure 7B*). Discriminating between these and other possibilities will require solving the challenge of analyzing purified NOCA-1 at physiological ionic strengths, which would enable better reconstitution of the interaction between NOCA-1 and microtubules (whether direct or indirect) in vitro and also enable analysis under conditions that include γ-tubulin-mediated nucleation.

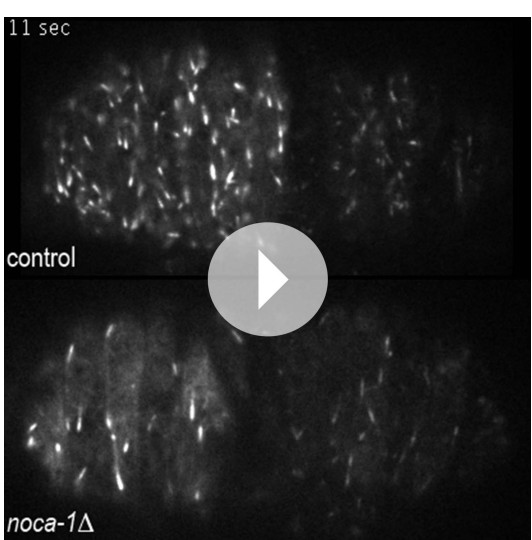

**Video 6.** Deletion of NOCA-1 reduces growing microtubules in the embryonic epidermis. Timelapse fluorescence confocal microscopy was used to acquire images of the dorsal epidermis in *C. elegans* embryos expressing EB1::GFP (P*lbp-1*:: EB1::GFP). Images of embryos from control and *noca-1(RNAi)* worms were acquired at 1 s intervals. Playback is 6× realtime.

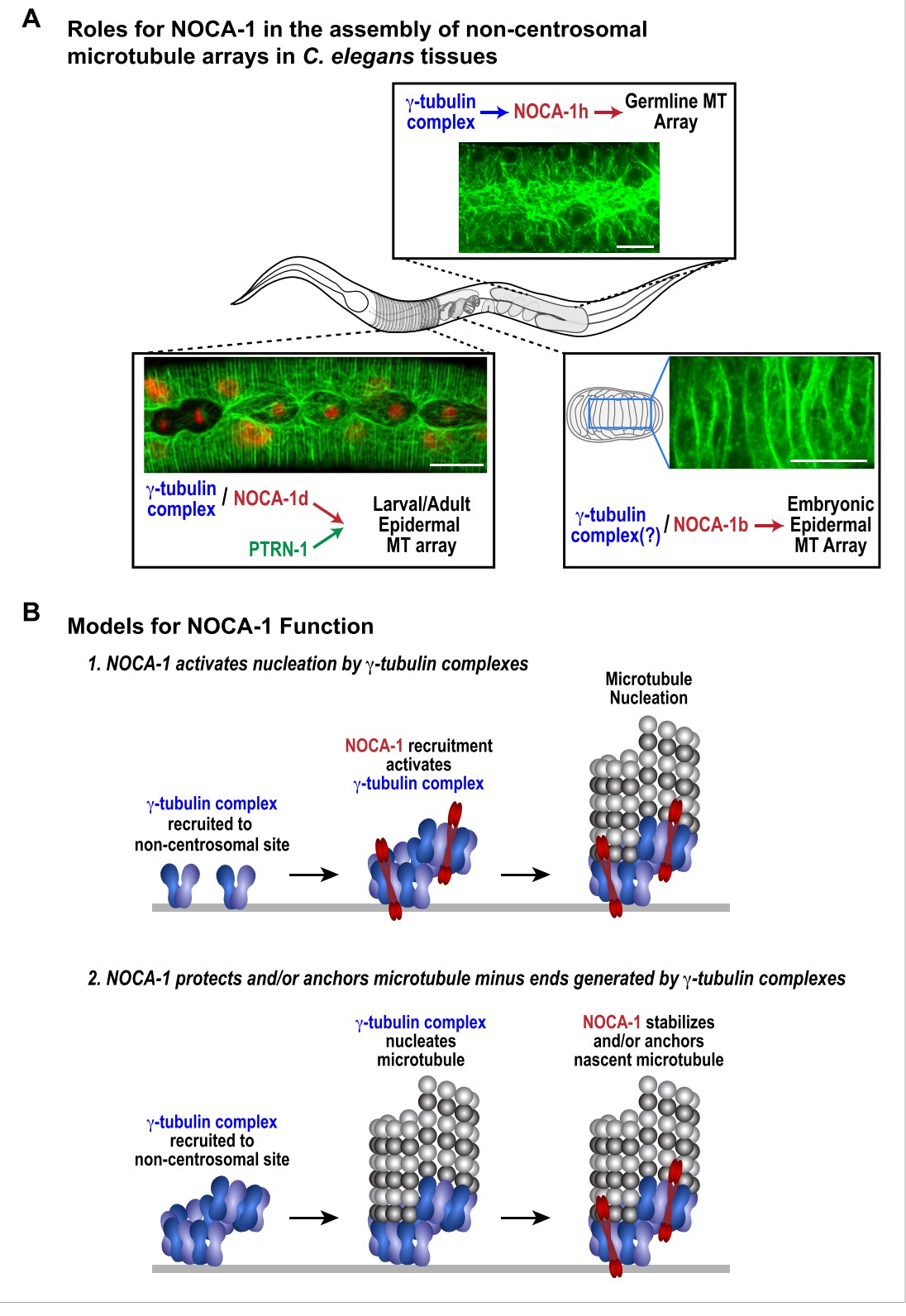

**Figure 7**. NOCA-1 functions in multiple *C. elegans* tissues to assemble non-centrosomal microtubule arrays. (**A**) Schematics and images summarizing the pathways that control the assembly of non-centrosomal microtubule arrays in three *C. elegans* tissues. (**B**) Schematics illustrating two speculative models for how NOCA-1 could function coordinately with the γ-tubulin complex to generate microtubule arrays. Scale bars, 10 μm.

## Relationship between NOCA-1 and Patronin/PTRN-1

NOCA-1 functions independently of PTRN-1 in some tissues, and in parallel to PTRN-1 in the larval/adult epidermis, where the NHD of NOCA-1 and PTRN-1 function redundantly to generate a circumferential array of microtubule bundles immediately juxtaposed to the plasma membrane (*Priess and Hirsh, 1986*; *Costa et al., 1997*). Imaging the dynamics of these bundles revealed that, despite the functional redundancy in supporting growth and morphogenesis, the microtubule arrays formed in the presence of NOCA-1 or PTRN-1 alone were distinct. When PTRN-1 was removed and only NOCA-1 was present, the microtubule growth rate and the number of growing EB1-marked microtubule ends were similar to controls. In contrast, removal of NOCA-1 led to a dramatic effect, causing a threefold reduction in the number of growing EB1-marked microtubule ends (*Figure 2D*). At the same time, the microtubules appeared to be less dynamic, and the appearance of the arrays combined with an ~20% reduction in growth rate suggests that there may be a small shift in the monomer/polymer balance towards more polymer. One possibility is that these effects result from the differences in the persistence of NOCA-1/γ-tubulin vs Patronin-based structures at microtubule minus ends. For example, NOCA-1/γ-tubulin might release microtubule minus ends more readily, perhaps leading to minus end depolymerization and shorter microtubules, whereas Patronin stretches might be less likely to be released leading to longer microtubules. Differences in microtubule length and minus-end dynamics could, in turn, affect plus-end dynamics. Alternatively, as has previously been proposed for γ-tubulin complexes (*Oakley et al., 2015*), it is possible that NOCA-1/γ-tubulin and Patronin-based structures affect plus-end dynamics by promoting the loading of different microtubule dynamicity factors. In this vein, the effect of NOCA-1 removal on EB1:: GFP localization is particularly interesting. When NOCA-1 is removed, increased amounts of EB1::GFP are observed along the length of the microtubules and an increase is also observed in EB1 comet length (*Figure 2D* and *Video 3*). It would be very interesting if NOCA-1/γ-tubulin vs Patronin-based structures at minus ends impacted the loading of factors that affect EB1 clearance from microtubules. The differences in the effects of NOCA-1 vs Patronin depletion raise the possibility that the choice between NOCA-1/ninein and/or PTRN-1/Patronin family members in different tissues may be related to the dynamicity (or lack thereof) required for the functions of different types of microtubule arrays. It will be particularly interesting to analyze NOCA-1 and PTRN-1 in the nervous system, where PTRN-1 has already been shown to support normal neuronal morphology and contribute to microtubule assembly and axon regeneration (*Chuang et al., 2014*; *Marcette et al., 2014*; *Richardson et al., 2014*).

The field is still in the early stages of investigating the question of redundancy between microtubule minus end-associated factors with respect to nucleating, stabilizing, and anchoring nascent minus ends. In vertebrate epithelial cells, Patronin/CAMSAP-mediated microtubule assembly has been reported to be independent of γ-tubulin-mediated nucleation and to potentially even compete with it (*Tanaka et al., 2012*). In contrast, in rat hippocampal neurons, γ-tubulin has been proposed to nucleate microtubules that are subsequently stabilized by CAMSAP2 (*Yau et al., 2014*). Our results in the *C. elegans* larval epidermis where NOCA-1 and Patronin/PTRN-1 are in parallel pathways with respect to microtubule generation, suggest that γ-tubulin cooperates with NOCA-1 but not with Patronin/PTRN-1. Whether Patronin/PTRN-1 promotes the assembly microtubules on its own in this context or functions together with other factors such as severing proteins (*Roll-Mecak and Vale, 2006*; *Lindeboom et al., 2013*) will be important to address in the future.

In summary, our work has shown that NOCA-1, a protein with homology to vertebrate ninein, functions together with γ-tubulin in the generation of microtubules in non-centrosomal microtubule arrays. Our results shed light on non-centrosomal microtubule array formation in diverse tissues in a whole organism and also reveal functional overlap between the ninein and Patronin families of microtubule cytoskeleton-regulating proteins.

## Materials and methods

### Worm strains

The *C. elegans* strains used in this study are listed in *Table 1*. All worm strains were maintained at 20°C on standard NGM plates seeded with OP-50 bacteria. The *noca-1(ok3692)* allele is balanced with a translocation balancer (nT1[qIs51]). However, as the *noca-1* locus is slightly outside of the balanced region (~2 cM from the translocation junction; *MacQueen et al., 2005*), worms containing nT1 balanced *noca-1(ok3692)* were maintained by singling individual worms at each generation from the

**Table 1.** *C. elegans* strains used in this study

| Strain # | Genotype |
|---|---|
| N2 | *wild type (ancestral)* |
| OD522 | *unc-119(ed3)III; ltSi62[pOD1110/pSW008; CEOP3608 TBG-1::mCherry; cb-unc-119(+)]III* |
| OD523 | *unc-119(ed3)III; ltSi63[pOD1111/pSW009; CEOP3608 TBG-1::GFP; cb-unc-119(+)]III* |
| OD528 | *unc-119(ed3)III; ttTi22935 V (Mos1 insertion)* |
| OD723 | *noca-1(ok3692)V/nT1[qIs51](IV;V)* |
| OD726 | *ltSi77[pOD1112/pSW032; Plbp-1::mCherry; cb-unc-119(+)]V* |
| OD747 | *unc-119(ed3) III; ttTi21011 X* |
| OD752 | *unc-119(ed3)III; ltSi182[pOD1237/pSW055; Pnoca-1::noca-1abcfgh; cb-unc-119(+)]III* |
| OD758 | *unc-119(ed3)III?; ltSi182[pOD1237/pSW055; Pnoca-1::noca-1abcfgh; cb-unc-119(+)]III; noca-1 (ok3692)V* |
| OD843 | *unc-119(ed3) III?; ltIs38 [pAA1; pie-1/GFP::PH(PLC1delta1); unc-119 (+)]; ltIs37 [pAA64; pie-1/ mCHERRY::his-58; unc-119 (+)] IV; noca-1(ok3692)V/nT1[qIs51](IV;V)* |
| OD851 | *unc-119(ed3) III?; ltSi62[pOD1110/pSW008; CEOP3608 TBG-1::mCherry; cb-unc-119(+)]III; noca-1(ok3692)V/nT1[qIs51](IV;V)* |
| OD854 | *ptrn-1(lt1::cb-unc-119+)X* |
| OD866 | *ltSi219[pOD1248/pSW076; Pmex-5::GFP::PH(PLC1delta1)::operon_linker::mCherry::his-11; cb-unc-119(+)]I* |
| OD868 | *ltSi220[pOD1249/pSW077; Pmex-5::GFP::tbb-2::operon_linker::mCherry::his-11; cb-unc-119(+)]I* |
| OD891 | *noca-1(ok3692)V/nT1[qIs51](IV;V); ptrn-1(lt1::cb-unc-119+)X* |
| OD907 | *ltSi222[pOD1250/pSW078; Plbp-1::GFP::tbb-2::operon_linker::mCherry::his-11; cb-unc-119(+)]I; noca-1(ok3692)V/nT1[qIs51](IV;V)* |
| OD909 | *ltSi222[pOD1250/pSW078; Plbp-1::GFP::tbb-2::operon_linker::mCherry::his-11; cb-unc-119(+)]I; ltSi77[pOD1112/pSW032; Plbp-1::mCherry; cb-unc-119(+)]V* |
| OD911 | *ltSi220[pOD1249/pSW077; Pmex-5::GFP::tbb-2::operon_linker::mCherry::his-11; cb-unc-119(+)] I; ptrn-1(lt1::cb-unc-119+)X* |
| OD952 | *unc-119(ed3)III; ltSi246[pOD1270/pSW082; Pnoca-1::noca-1abcfgh::superfolderGFP; cb-unc-119 (+)]III* |
| OD961 | *ltSi249[pOD1274/pSW098; Pdlg-1delta7::dlg-1::GFP::unc-54-3' UTR; cb-unc-119(+)]I* |
| OD1011 | *ltSi220[pOD1249/pSW077; Pmex-5::GFP::tbb-2::operon_linker::mCherry::his-11; cb-unc-119(+)] I; noca-1(ok3692)V/nT1(IV;V)* |
| OD1222 | *ltSi182[pOD1237/pSW055; Pnoca-1::noca-1abcfgh; cb-unc-119(+)]III; unc-119(ed3)III?; noca-1 (ok3692)V; ptrn-1(lt1::cb-unc-119+)X* |
| OD1223 | *unc-119(ed3)III; ltSi364[pOD1330/pSW147; Pnoca-1h::noca-1h(1-251)::superfolderGFP; cb-unc-119(+)]III* |
| OD1225 | *unc-119(ed3)III; ltSi366[pOD1332/pSW149; Pnoca-1h::noca-1h(457-922)::superfolderGFP; cb-unc-119(+)]III* |
| OD1227 | *unc-119(ed3)III; ltSi368[pOD1334/pSW151; Pnoca-1h::noca-1h(252-922)::superfolderGFP; cb-unc-119(+)]III* |
| OD1233 | *ltSi369[pOD1335/pSW152; Pnoca-1h::noca-1h(RNAi resistant)::superfolderGFP; cb-unc-119(+)]III* |
| OD1339 | *unc-119(ed3)III; ltSi417[pOD1342/pSW159; Pnoca-1de::noca-1de::mCherry; cb-unc-119(+)]III* |
| OD1345 | *ltSi417[pOD1342/pSW159; Pnoca-1de::noca-1de::mCherry; cb-unc-119(+)]III; unc-119(ed3)III?; noca-1(ok3692)V/nT1[qIs51](IV;V); ptrn-1(lt1::cb-unc-119+)X* |
| OD1347 | *ltSi419[pOD1465/pSW177; Pnoca-1h::ptrn-1(cDNA)::superfolderGFP; cb-unc-119(+)]III; unc-119 (ed3)III* |
| OD1359 | *ltSi716[pOD1935/pDC208; Pmex-5::EBP-2::GFP::tbb-2_3' UTR; cb-unc-119(+)]I; unc-119(ed3)III* |
| OD1394 | *ltSi443[pOD1471/pSW182; Pnoca-1h::noca-1h(1-251)::superfolderGFP (C10A); cb-unc-119(+)]III; unc-119(ed3)III* |
| OD1426 | *ltSi449[pOD1461/pSW173; Plbp-1::EBP-2::GFP::opLinker::mCherry::PH; cb-unc-119(+)]I; unc-119(ed3)III* |

*Table 1. Continued on next page*

Table 1. Continued

| Strain # | Genotype |
| --- | --- |
| OD1442 | ltSi458[pOD1477/pSW188; Pnoca-1d::noca-1d(cDNA)::mCherry; cb-unc-119(+)]II; unc-119 (ed3)III |
| OD1443 | ltSi459[pOD1478/pSW189; Pnoca-1e::noca-1e(cDNA)::mCherry; cb-unc-119(+)]II; unc-119(ed3)III |
| OD1446 | ltSi461[pOD1340/pSW157; Pnoca-1::noca-1abc*gh (STOP in the first exon of isoform f); cb-unc-119(+)]II; unc-119(ed3)III |
| OD1504 | ltSi449[pOD1461/pSW173; Plbp-1::EBP-2::GFP::opLinker::mCHerry::PH; cb-unc-119(+)]I; unc-119(ed3)?III; noca-1(ok3692)V/nT1[qIs51](IV;V) |
| OD1505 | ltSi449[pOD1461/pSW173; Plbp-1::EBP-2::GFP::opLinker::mCHerry::PH; cb-unc-119(+)]I; unc-119(ed3)?III; ltSi77[pOD1112/pSW032; Plbp-1::mCherry; cb-unc-119(+)]V |
| OD1510 | ltSi249[pOD1274/pSW098; Pdlg-1delta7::dlg-1::GFP::unc-54-3' UTR; cb-unc-119(+)]I; noca-1(ok3692)V/nT1[qIs51](IV;V) |
| OD1511 | ltSi249[pOD1274/pSW098; Pdlg-1delta7::dlg-1::GFP::unc-54-3' UTR; cb-unc-119(+)]I; ptrn-1(lt1::cb-unc-119+)X |
| OD1512 | ltSi249[pOD1274/pSW098; Pdlg-1delta7::dlg-1::GFP::unc-54-3' UTR; cb-unc-119(+)]I; noca-1(ok3692)V/nT1[qIs51](IV;V); ptrn-1(lt1::cb-unc-119+)X |
| OD1516 | ltSi458[pOD1477/pSW188, Pnoca-1d::noca-1d(cDNA)::mCherry; cb-unc-119(+)]II; unc-119(ed3)?III; noca-1(ok3692)V/nT1[qIs51](IV;V); ptrn-1(lt1::cb-unc-119+)X |
| OD1517 | ltSi459[pOD1478/pSW189; Pnoca-1e::noca-1e(cDNA)::mCherry; cb-unc-119(+)]II; unc-119(ed3)?III; noca-1(ok3692)V/nT1[qIs51](IV;V); ptrn-1(lt1::cb-unc-119+)X |
| OD1521 | ltSi461[pOD1340/pSW157; Pnoca-1::noca-1abc*gh (STOP in the first exon of isoform f); cb-unc-119(+)]II; unc-119(ed3)?III; noca-1(ok3692)V |
| OD1558 | ltSi518[pOD1338/pSW155; Pnoca-1::noca-1a*cfgh(STOP coden in the first exon of isoform b); cb-unc-119(+)]II; unc-119(ed3)III |
| OD1578 | ltSi523[pOD1339/pSW156; Pnoca-1::noca-1ab*fgh(STOP coden in the first exon of isoform c); cb-unc-119(+)]II; unc-119(ed3)III |
| OD1580 | ltSi518[pOD1338/pSW155; Pnoca-1::noca-1a*cfgh(STOP coden in the first exon of isoform b); cb-unc-119(+)]II; unc-119(ed3)III?; noca-1(ok3692)V |
| OD1600 | ltSi523[pOD1339/pSW156; Pnoca-1::noca-1ab*fgh(STOP coden in the first exon of isoform c); cb-unc-119(+)]II; unc-119(ed3)III?; noca-1(ok3692)V |
| OD1605 | ltSi531[pOD1337/pSW154; Pnoca-1::noca-1*bcfgh(STOP coden in the first exon of isoform a); cb-unc-119(+)]II; unc-119(ed3)III |
| OD1606 | ltSi531[pOD1337/pSW154; Pnoca-1::noca-1*bcfgh(STOP coden in the first exon of isoform a); cb-unc-119(+)]II; unc-119(ed3)III?; noca-1(ok3692)V |
| OD1652 | ltSi540[pOD1343/pSW160; Pnoca-1de::noca-1de::superfolderGFP; cb-unc-119(+)]II; unc-119(ed3)III |
| OD1653 | ltSi541[pOD1505/pSW210; Pdpy-7::PTRN-1(cDNA)::superfolderGFP; cb-unc-119(+)]II; unc-119(ed3)III |
| OD1654 | ltSi542[pOD1506/pSW211; Pptrn-1::PTRN-1(cDNA)::superfolderGFP; cb-unc-119(+)]II; unc-119(ed3)III |
| OD1690 | ltSi561[pOD1508/pSW213; Pptrn-1::noca-1h(457-922)::superfolderGFP; cb-unc-119(+)]II; unc-119(ed3)III |
| OD1691 | ltSi562[pOD1509/pSW214; Pptrn-1::noca-1h(252-922)::superfolderGFP; cb-unc-119(+)]II; unc-119(ed3)III |
| OD1708 | ltSi568[pOD1518/pSW223; Pmex-5::mCherry::PH::tbb-2_3' UTR; cb-unc-119(+)]I; unc-119(ed3)III |
| OD1709 | ltSi569[oxTi185; pOD1110/pSW008; CEOP3608 TBG-1::mCherry; cb-unc-119(+)]I; unc-119 (ed3)III |
| OD1727 | ltSi569[oxTi185; pOD1110/pSW008; CEOP3608 TBG-1::mCherry; cb-unc-119(+)]I; ltSi246 [pOD1270/pSW082; Pnoca-1::noca-1abcfgh::superfolderGFP; cb-unc-119(+)]II; unc-119(ed3)III? |
| OD1731 | ltSi568[pOD1518/pSW223; Pmex-5::mCherry::PH::tbb-2_3' UTR; cb-unc-119(+)]I; ltSi369 [pOD1335/pSW152; Pnoca-1h::noca-1hRR::superfolderGFP; cb-unc-119(+)]II; unc-119(ed3)III? |
| OD1737 | ltSi542[pOD1506/pSW211; Pptrn-1::PTRN-1(cDNA)::superfolderGFP; cb-unc-119(+)]II; unc-119 (ed3)III?; noca-1(ok3692)V/nT1[qIs51](IV;V); ptrn-1(lt1::cb-unc-119+)X |
| OD1739 | ltSi561[pOD1508/pSW213; Pptrn-1::noca-1h(457-922)::superfolderGFP; cb-unc-119(+)]II; unc-119(ed3)III?; noca-1(ok3692)V/nT1[qIs51](IV;V); ptrn-1(lt1::cb-unc-119+)X |

Table 1. Continued on next page

Table 1. Continued

| Strain # | Genotype |
| --- | --- |
| OD1740 | ltSi562[pOD1509/pSW214; Pptrn-1::noca-1h(252-922)::superfolderGFP; cb-unc-119(+)]III; unc-119(ed3)III?; noca-1(ok3692)V/nT1[qIs51](IV;V); ptrn-1(lt1::cb-unc-119+)X |
| OD1741 | ltSi570[pOD1527/pSW232; Pdpy-7::GFP::tbb-2::mCHerry::his-11; cb-unc-119(+)]I; unc-119 (ed3)III |
| OD1742 | ltSi419[pOD1465/pSW177; Pnoca-1h::ptrn-1(cDNA)::superfolderGFP; cb-unc-119(+)]III; unc-119 (ed3)III?; noca-1(ok3692)V/nT1[qIs51](IV;V) |
| OD1780 | ltSi570[pOD1527/pSW232; Pdpy-7::GFP::tbb-2::mCHerry::his-11; cb-unc-119(+)]I; unc-119(ed3) III?; noca-1(ok3692)V/nT1[qIs51](IV;V) |
| OD1781 | ltSi570[pOD1527/pSW232; Pdpy-7::GFP::tbb-2::mCHerry::his-11; cb-unc-119(+)]I; unc-119(ed3) III?; ptrn-1(lt1::cb-unc-119+)X |
| OD1782 | ltSi570[pOD1527/pSW232; Pdpy-7::GFP::tbb-2::mCHerry::his-11; cb-unc-119(+)]I; unc-119(ed3) III?; noca-1(ok3692)V/nT1[qIs51](IV;V); ptrn-1(lt1::cb-unc-119+)X |
| OD1864 | ltSi598[pOD1553/pSW252; Plbp-1::noca-1b::superfolderGFP::opLinker::mCHerry::PH; cb-unc-119(+)]III; unc-119(ed3)III |
| OD1865 | ltSi599[pOD1554/pSW253; Plbp-1::noca-1h(252-922)::superfolderGFP::opLinker::mCHerry::PH; cb-unc-119(+)]III; unc-119(ed3)III |
| OD1866 | ltSi600[pOD1555/pSW254; Plbp-1::noca-1h(457-922)::superfolderGFP::opLinker::mCHerry::PH; cb-unc-119(+)]III; unc-119(ed3)III |
| OD1867 | ltSi601[pOD1542/pSW244; Ppha-4int1::PTRN-1(cDNA)::superfolderGFP; cb-unc-119(+)]III; unc-119(ed3)III |
| OD1869 | ltSi603[pOD1544/pSW246; Prgef-1::PTRN-1(cDNA)::superfolderGFP; cb-unc-119(+)]III; unc-119 (ed3)III |
| OD1908 | ltSi598[pOD1553/pSW252; Plbp-1::noca-1b::superfolderGFP::opLinker::mCHerry::histone; cb-unc-119(+)]III; unc-119(ed3)III?; noca-1(ok3692)V/nT1[qIs51](IV;V) |
| OD1909 | ltSi599[pOD1554/pSW253; Plbp-1::noca-1h(252-922)::superfolderGFP::opLinker::mCHerry:: histone; cb-unc-119(+)]III; unc-119(ed3)III?; noca-1(ok3692)V/nT1[qIs51](IV;V) |
| OD1910 | ltSi600[pOD1555/pSW254; Plbp-1::noca-1h(457-922)::superfolderGFP::opLinker::mCHerry:: histone; cb-unc-119(+)]III; unc-119(ed3)III?; noca-1(ok3692)V/nT1[qIs51](IV;V) |
| OD1911 | ltSi601[pOD1542/pSW244; Ppha-4int1::PTRN-1(cDNA)::superfolderGFP; cb-unc-119(+)]III; unc-119(ed3)III?; noca-1(ok3692)V/nT1[qIs51](IV;V); ptrn-1(lt1::cb-unc-119+)X |
| OD1913 | ltSi603[pOD1544/pSW246; Prgef-1::PTRN-1(cDNA)::superfolderGFP; cb-unc-119(+)]III; unc-119 (ed3)III?; noca-1(ok3692)V/nT1[qIs51](IV;V); ptrn-1(lt1::cb-unc-119+)X |
| OD1914 | ltSi219[pOD1248/pSW076; Pmex-5::GFP::PH(PLC1delta1)::operon_linker::mCHerry::his-11; cb-unc-119(+)]I; ptrn-1(lt1::cb-unc-119+)X |
| OD2006 | ltSi541[pOD1505/pSW210; Pdpy-7::PTRN-1(cDNA)::superfolderGFP; cb-unc-119(+)]III; unc-119 (ed3)III?; noca-1(ok3692)V/nT1[qIs51](IV;V); ptrn-1(lt1::cb-unc-119+)X |
| OD2074 | ltSi670[pSW268/pOD1786; Pmex-5::noca-1h(252-922)::superfolderGFP::opLinker::mCHerry::PH; cb-unc-119(+)]I; unc-119(ed3)III |
| OD2111 | ltSi673[pSW279/pOD1787; Pdpy-7::tagRFP::tbb-2; cb-unc-119(+)]I; unc-119(ed3)III |
| OD2113 | ltSi673[pSW279/pOD1787; Pdpy-7::tagRFP::tbb-2; cb-unc-119(+)]I; ltSi540[pOD1343/pSW160; Pnoca-1de::noca-1de::superfolderGFP; cb-unc-119(+)]III; unc-119(ed3)III? |
| OD2114 | ltSi673[pSW279/pOD1787; Pdpy-7::tagRFP::tbb-2; cb-unc-119(+)]I; ltSi542[pOD1506/pSW211; Pptrn-1::PTRN-1(cDNA)::superfolderGFP; cb-unc-119(+)]III; unc-119(ed3)III |
| OD2115 | ltSi569[oxTi185; pOD1110/pSW008; CEOP3608 TBG-1::mCherry; cb-unc-119(+)]I; ltSi540 [pOD1343/pSW160; Pnoca-1de::noca-1de::superfolderGFP; cb-unc-119(+)]III; unc-119(ed3)III? |
| OD2116 | ltSi569[oxTi185; pOD1110/pSW008; CEOP3608 TBG-1::mCherry; cb-unc-119(+)]I; ltSi542 [pOD1506/pSW211; Pptrn-1::PTRN-1(cDNA)::superfolderGFP; cb-unc-119(+)]III; unc-119(ed3)III? |
| OD2396 | mcIs46[pCL08(dlg-1::RFP); cb-unc-119(+)]?; mcSi53[Pdpy-7::EB1::GFP; cb-unc-119(+)]III; noca-1 (ok3692)V/nT1[qIs51](IV;V); |
| OD2397 | mcIs46[pCL08(dlg-1::RFP); cb-unc-119(+)]?; mcSi53[Pdpy-7::EB1::GFP; cb-unc-119(+)]III; ptrn-1 (lt1::cb-unc-119+)X |
| OD2435 | ltSi569[oxTi185; pOD1110/pSW008; CEOP3608 TBG-1::mCherry; cb-unc-119(+)]I; ltSi202 [pVV103; Pspd-2::GFP::SPD-5 reencoded; cb-unc-119(+)]III; unc-119(ed3) III |

Table 1. Continued on next page

Table 1. Continued

| Strain # | Genotype |
|---|---|
| OD2442 | ltSi794[pOD1988/pSW302; Pdpy-7::vhhGFP4::ZIF-1::unc-54_3′ UTR; cb-unc-119(+)]II; unc-119 (ed3)III |
| OD2509 | gip-2(lt19[gip-2::GFP]::loxP::cb-unc-119(+)::loxP]I; unc-119(ed3)III |
| OD2624 | gip-2(lt19[gip-2::GFP]::loxP::cb-unc-119(+)::loxP]I; ltSi794[pOD1988/pSW302; Pdpy-7::vhhGFP4:: ZIF-1::unc-54_3′ UTR; cb-unc-119(+)]II; unc-119(ed3)III?; noca-1(ok3692)V/nT1[qIs51](IV;V) |
| OD2625 | gip-2(lt19[gip-2::GFP]::loxP::cb-unc-119(+)::loxP]I/hT2[bli-4(e937) let-?(q782) qIs48](I;III); ltSi794 [pOD1988/pSW302; Pdpy-7::vhhGFP4::ZIF-1::unc-54_3′ UTR; cb-unc-119(+)]II; unc-119(ed3)III?; ptrn-1(lt1::cb-unc-119+)X |
| OD2626 | gip-2(lt19[gip-2::GFP]::loxP::cb-unc-119(+)::loxP]I; ltSi794[pOD1988/pSW302; Pdpy-7::vhhGFP4:: ZIF-1::unc-54_3′ UTR; cb-unc-119(+)]II; unc-119(ed3)III?; ptrn-1(lt1::cb-unc-119+)X |
| ML1654 | mcIs46[pCL08(dlg-1::RFP); cb-unc-119(+)]?; mcSi53[Pdpy-7::EB1::GFP; cb-unc-119(+)]II |

progeny of worms yielding the proper phenotypic distribution (4/5 fertile worms with pharyngeal GFP and 1/5 sterile worms without pharyngeal GFP).

A transposon-based deletion strategy (MosDEL; *Frøkjær-Jensen et al., 2010*) was used to make the null *ptrn-1Δ* allele (*ptrn-1(lt1::cb-unc-119+)*; *Figure 1—figure supplement 3*). Briefly, a repair plasmid containing the *Cb-unc-119* selection marker and appropriate homology arms (pOD1877, 50 ng/µl) was co-injected with a plasmid encoding the Mos1 transposase (pJL43.1, P*glh-2::Mos1 transposase*, 50 ng/µl) and three plasmids encoding fluorescent markers for negative selection (pCFJ90 [P*myo-2::mCherry*, 2.5 ng/µl], pCFJ104 [P*myo-3::mCherry*, 5 ng/µl] and pGH8 [P*rab-3::mCherry*, 10 ng/µl]) into the strain OD747. After 1 week, moving progeny lacking fluorescent markers were identified and *ptrn-1* deletion was confirmed in their progeny by PCR spanning both homology regions.

A similar transposon-based strategy (MosSCI; *Frøkjær-Jensen et al., 2008*) was used to generate all of the transgenes used in this study. To make the *noca-1h::superfolderGFP* (superfolder GFP is a folding-improved GFP version; see *Pédelacq et al., 2006*) transgene RNAi resistant, a 999-bp region close to the 3′-end of the *noca-1* coding sequence was re-encoded by codon shuffling (*Figure 4—figure supplement 2*). Depending on which Mos1 insertion site was used, transgenes were cloned into pCFJ151 (ChrII insertion, *ttTi5605*; UniI insertion, *oxTi185*; UniIV insertion, *oxTi177*), pCFJ352 (ChrI insertion, *ttTi4348*), or were cloned de novo (assembly of multiple linear DNA fragments obtained by PCR [*Gibson et al., 2009*]; ChrV insertion, *ttTi22935*). In most cases, an improved transposase plasmid using a stronger promoter (pCFJ601, P*eft-3::Mos1 transposase*, 50 ng/µl) and an additional negative selection marker pMA122 (P*hsp-16.41::peel-1*, 10 ng/µl) were used in the injection mix. Single copy transgenes were generated by injecting a mixture of repairing plasmid, transposase plasmid, and selection markers into strains EG6429 (*ttTi5605*, Chr II), EG6701 (*ttTi4348*, Chr I), EG8078 (*oxTi185*, Chr I), or EG8081 (*oxTi177*, Chr IV). After 1 week, progeny of injected worms were heat shocked at 34°C for 2–4 hr to induce the expression of PEEL-1, in order to kill extra chromosomal array containing worms (*Seidel et al., 2011*). Moving worms without fluorescent markers were identified and transgene integration was confirmed in their progeny by PCR spanning both homology regions.

A CRISPR/Cas9-based method (*Dickinson et al., 2013*) was used to generate the endogenously tagged *gip-2::GFP* strain. Briefly, a repairing plasmid containing the *Cb-unc-119* selection marker and appropriate homology arms (678 bp at the 3′-end of *gip-2* coding sequence and 750 bp for the *gip-2* 3′ UTR; pOD1999, 20 ng/µl) was co-injected with two plasmids modified from pDD162 by inserting two different guide RNA sequences (5′-AGTTCAGTCAAGAGCTCGAA-3′ and 5′-TTATTATGTCTTTTGGGTAT-3′; the plasmid also encodes the Cas9 protein; 50 ng/µl for each), three plasmids encoding fluorescent markers for negative selection (pCFJ90 [P*myo-2::mCherry*, 2.5 ng/µl], pCFJ104 [P*myo-3::mCherry*, 5 ng/µl] and pGH8 [P*rab-3:: mCherry*, 10 ng/µl]) and one plasmid encoding a heat shock-inducible toxin (pMA122, P*hsp-16.41::peel-1*, 10 ng/µl) into the strain HT1593. After 1 week, progeny of injected worms were heat shocked at 34°C for 2 hr to induce the expression of PEEL-1, in order to kill extra chromosomal array containing worms (*Seidel et al., 2011*). Moving worms without fluorescent markers were identified and GFP insertion was confirmed in their progeny by PCR spanning both homology regions.

Homozygous *noca-1Δ* embryos (**Figure 6A,B**) were obtained from heterozygous *noca-1Δ* mothers, because *noca-1Δ* mutants are completely sterile. To distinguish the homozygous *noca-1Δ* embryos from its wild-type and heterozygous siblings, we used MosSCI (**Frøkjær-Jensen et al., 2008**) to insert a reporter transgene (cytoplasmic mCherry driven by an epidermal promoter P*lbp-1*) at a site only 0.06 cM from the *noca-1* locus, so that the wild-type *noca-1+* is tightly linked to the mCherry reporter. To obtain *noca-1Δ* embryos, the reporter strain was mated with the *nT1[qls51]* balanced *noca-1Δ* strain (both strains expressing GFP::β-tubulin in the embryonic epidermis) to get heterozygous P*lbp-1::mCherry(noca-1+)/noca-1Δ* worms. Embryos from these worms were dissected out, mounted on 2% agarose pad and imaged (see 'Light microscopy' section).

## Identification of NOCA-1 isoform h

Seven splicing isoforms of *noca-1* have been annotated in Wormbase. We noticed that two EST clones (yk322h12 and yk639c8, Wormbase) spanned *noca-1* and its 5′ neighboring gene K03H4.2, suggesting the existence of an additional previously unannotated *noca-1* isoform. 5′-RACE (Invitrogen, Carlsbad, CA; 18374-058) using three gene-specific primers: 5′-gcttccattgaaatgagacgat-3′, 5′-gacgaagaatgtctcgactgg-3′ and 5′-tggcttggtgttgaatgaga-3′ (within the exons shared by all known isoforms) revealed an eighth isoform (*noca-1h*). The sequence of full-length *noca-1h* amplified from *C. elegans* cDNA is inserted below.

>*noca-1*, isoform h

atgctcaaacaactattggctttgacttgcatgcacaaaaaagataaaaataagcttgcaataactgctggaaccgcagaatgttcgaacagatctcctcaaaattcaccgggatcttcctctgaaggcgctgcagacgaatctctaaatcagagtgttgctattccggaagaagctcatctgaacacttcacagtttatttcacttcccctctccgacgtctcatttgaagccgctgcatctcaaaatcgagctacaccgattgattttggtacacgagaagtgaaagaagatgacgatgttctcagtgacactggtcgtcgtcgaagcgttaacttaataacgccttctcctattccagaagaaaccgaggataacttaacagaaacgcctattcctgtagttgaacacattccaagaagtgcaatttttgaacctttcaatcacgaaaattctcctttgttctccgtgaaggcacgtaagaaagctcatgaataccgctccaacgattcaactctcagtccttcatcatcttccaacaatgacgacagtatccggattgacagtatccgtgtccgttcatcaaaatctgcaacgaataatcaactgaaaggacggcttacaccaatactaggagggtcccttcgcccgattccaaaaaaaaggaaccgagtcgctttcaacggaaattctacatttgtcgcaccggagagactatgcttggaagttgataaaatacatcaagatcgtttcgcctccgtaaacgtggagatacgtcccgtcgagatgcagtggaagctggattcgaaccgagagatactgttccacgatgtcattcaacacagtcgttgagagatgttcaacgtgttcgatcatacaacaattcacagtttcaggccagtgatctttcactcaatccaaatggaagtattcgtgctgcttgtgacagtacaagtggatctgtcgccccaacagcagttgtaaatcctgcccggaatcatgtcatttcacatcgacaacaacatcatacaagctacgagaaggatcttattccccatcataacattgatgtggatcgtcgccgtagtttgcaagctctcaatggctcatctgctctctatcaactaaataatggcggttcaccgaatggagtgagatctcaatttttccttcggatctttctatccatacaccagttcatcatgttggaagtcgagttcgagtgtccagtgtcaaccagatttgcgattcgaacagtgctccacaattcagtatcgatcaacgccgcagtgttcacaacattggaaatccggttcgaaattcgtttgtggatggaataaaaactacatcgactccaaaaaatcagatagcagttgctccactggctcacaaaagtagacatttgagtgaatctcgagatgagatgcgtggcggtgcagaacgacgtggcagtggtggtcaaatgaatttaccagcctacactaattatcttatacgccattctggagaagagcgtcttgtggatggaccggtcactaatgccagcgatgctcggattgcttatcttgaaaaacgaatccgagaacttgaactgacacaaaaagaacagagctctcattcaacaccaagccagtcgagacattcttcgtctaaatcgtctcatttcaatggaagcagtaacttgtctacaagcgaacaactccgattgcaagaaatgagcgatgagttggcgaacaaggatcgtaaagttacatctttggaatcgaagcttctgaaagcttatcaaagaattgaacgactgaacgaggagtacgacggaaaaataaaaaatctgatgtatgacagtgaacgtgctcgcgacgatctcactcgatgtgttgataagattcagcaattggaaaacgaacttgatgagacacgagctgcagtacaaaatggagatcatgcaaatgaacaggaatatcatgagttacgagataagatctggaaacaagaacgtgaacttcaagagagtcgtacgttgcttactcgtttgcgagaaaaagaagcagaatttgagagaatgcgatcagagaaaggatatcttgagttgaagaatgagaatctcaacaagaaattggaagcgaaaaagcgagcagttgaagaactcgaaagaagtgtttcgactcttcgattggagcaaactatttgccagcaatcatgctcatctggatcaacaccgcttgctgatgagatggagattatgtcagatatccgaccatcactcgccagaccatacaccaaggctcattcgacactcgggtcccacaatatgtcaccactatcgcactcaaagtccagtggattaacgaagagtttttcgaattttgcgctcaactcatctaaacagcgtgatgatatcaccgccaatatgagccgatcgattcgtgaacaaaaccgtcacataacaatgtgtagagctatggttgtttgtctgaaggatacggtagaccgaatggcacgtggagagaatcctgatgttgctcgtctgctcggtgtcaagttgaatgtgatgtctgaaagtgaaatggaagatgatgaagatcatgaggctgatgcatcacaaccgttttcaatgatgtctgctgaatcagcgctctcgaagcaatgcggaaaactcgctgatctcgataaagacctagatacaattcgctgtcaactcgcagattggcatggtcaaacaaatgcagaaggagatggtgatcgtgatgtatgcagagttcaatag

## Palmitoylation predication

CSS-Palm 4.0 with the default medium threshold (**Ren et al., 2008**; http://csspalm.biocuckoo.org/) identified Cysteine 10 of NOCA-1h as a putative palmitoylation site (score 38.139 vs the cut-off score of 3.717).

## RNA interference

Single-stranded RNAs (ssRNAs) were synthesized in 50 µl T3 and T7 reactions (MEGAscript, Invitrogen) using cleaned DNA templates generated by PCR from N2 cDNA using the oligos in *Table 2*. Reactions were cleaned using the MEGAclear kit (Invitrogen), and the 50 µl T3 and T7 reactions were mixed with 50 µl of 3× soaking buffer (32.7 mM $Na_2HPO_4$, 16.5 mM $KH_2PO_4$, 6.3 mM NaCl, 14.1 mM $NH_4Cl$), denatured at 68°C for 10 min, and then annealed at 37°C for 30 min to generate dsRNA.

For localization analysis in *Figure 5D,E* and cell division analysis in *Figure 2—figure supplement 1*, dsRNA was delivered by injecting L4 hermaphrodites. For all other RNAi experiments, dsRNA was delivered by soaking L4 hermaphrodites for 24 hr at 20°C. After recovery from injection or soaking, worms were incubated at 20°C for 18–54 hr before different experiments. For brood size counting and embryonic lethality assays, worms were singled 24 hr post recovery and removed from the plates at 48 hr post recovery. The number of hatched larvae and unhatched embryos were counted 1 day later. For germline imaging, injected worms were incubated at 20°C for 48–54 hr before imaging.

## *C. elegans* assays

For larval lethality assays, embryos were obtained by bleaching adult worms with freshly mixed 20% household bleach and 0.5 N NaOH for 10 min. Embryos were then rinsed twice in M9 and rotated in M9 at room temperature (~23°C) overnight to allow hatching. On the following day, synchronized, starved L1 worms were recovered on food. Phenotypes were quantified 72 hr post recovery.

For the permeability assay (*Moribe et al., 2004*), synchronized worms were rinsed with M9 in a depression slide well and transferred into 1 µg/ml HOECHST 33258 (Sigma–Aldrich, St. Louis, MO) in M9. After 15 min, worms were rinsed twice in M9 and anesthetized in M9 with 1 mg/ml Tricaine (ethyl 3-aminobenzoate methanesulfonate salt, Sigma–Aldrich; A5040-25G) and 0.1 mg/ml TMHC (tetramisole hydrochloride, Sigma–Aldrich; T1512-10G) for 30 min. Worms were mounted onto a 2% agarose pad and imaged on a Nikon eclipse E800 microscope (see 'Light microscopy' section).

For brood size and embryonic lethality assays, L4 hermaphrodites were incubated at 20°C and singled 24 hr later. After another 24 hr, the adult worms were removed from the plates. Embryos were allowed to develop for 20–30 hr and hatched and unhatched (embryonic lethal) worms were counted the following day.

For nuclear migration assays, worms were maintained at 20°C. Healthy L1 worms were partially anesthetized in 20 mM $NaN_3$, mounted on a 2% agarose pad, and imaged using DIC optics on an inverted Zeiss Axio Observer Z1 microscope system (see 'Light microscopy' section) and the number of nuclei in the dorsal cord was counted.

## Light microscopy

Images and videos in *Figures 2B–E, 3A,B,D, 4A,C,D, 6A–C, 7A*, *Figure 2—figure supplement 2*, *Figure 2—figure supplement 4*, *Figure 2—figure supplement 6*, *Figure 3—figure supplement 1*, *Figure 4—figure supplement 1A* (control and *noca-1(RNAi)*), *Videos 2, 3, 5, 6* were acquired using an inverted Zeiss Axio Observer Z1 system with a Yokogawa spinning-disk confocal head (CSU-X1), a 63× 1.40 NA Plan Apochromat lens (Zeiss, Oberkochen, Germany), and a Hamamatsu ORCA-ER camera (Model C4742-95-12ERG, Hamamatsu photonics, Shizuoka, Japan). Images in *Figures 1G, 4B,E, 5A–5C*, *Figure 1—figure supplement 5A*, *Figure 2—figure supplement 1A*, *Figure 4—figure*

**Table 2**. Oligos used for dsRNA production

| Gene | Oligonucleotide 1 | Oligonucleotide 2 | Template | mg/ml |
|---|---|---|---|---|
| T09E8.1 (*noca-1*) | AATTAACCCTCACTAAAGG ggcgaacaaggatcgtaaag | TAATACGACTCACTATAGG ctgcatttgtttgaccatgc | N2 cDNA | 1.8 |
| T09E8.1h (*noca-1h*) | AATTAACCCTCACTAAAGG gcttgcaataactgctggaa | TAATACGACTCACTATAGG aagcgactcggttcctttt | N2 cDNA | 1.1 |
| F58A4.8 (*tbg-1*) | AATTAACCCTCACTAAAGG ctcaagccttctggaaatcg | TAATACGACTCACTATAGG ccatgctcttcagcaacg | N2 cDNA | 1.1 |
| F26E4.8 (*tba-1*) | AATTAACCCTCACTAAAGG ccgatactggaaacggaaga | TAATACGACTCACTATAGG tggtgtaacttggacggtca | N2 cDNA | 1.5 |

supplement 1A (*noca-1Δ*), *Figure 4—figure supplement 2B* and *Figure 5—figure supplement 1A* were acquired using the same system with an EMCCD camera (QuantEM:512SC, Photometrics, Tucson, AZ). Imaging parameters were controlled using AxioVision software (Zeiss).

Images in *Figure 5D–F, 6D* were acquired using a Nikon TE2000-E inverted microscope with a Yokogawa spinning-disk confocal head (CSU-10), a 60× 1.40 NA Plan Apochromat lens (Nikon, Tokyo, Japan) and an EMCCD camera (iXon DV887ECS-BV, Andor Technology, Belfast, United Kingdom). Imaging parameters were controlled using the Andor iQ2 software. Images in *Figure 1—figure supplement 5B* was acquired using the same system as above except that a 100× 1.40 NA Plan Apochromat lens (Nikon) was used.

Images in *Figure 2F* and *Figure 2—figure supplement 5* were acquired using a Nikon eclipse E800 microscope with a 60× 1.40 NA Plan Apo lens (Nikon) and a Hamamatsu ORCA-ER camera (Model C4742-95-12ERG, Hamamatsu photonics). Imaging parameters were controlled using the Metamorph software (Molecular Devices, Sunnyvale, CA).

Images and videos in *Figures 1D, 3E*, *Videos 1, 4* were acquired using the DinoEye eyepiece camera (AM7023B, Dino-Lite, Hsinchu, Taiwan) mounted on a Nikon SMZ800 dissection scope. Imaging parameters were controlled using the DinoXcope software (Dino-Lite).

## Image analysis

EB1 comet counting was performed using imageJ software (FIJI) in a macro-aided semi-automatic fashion. Tiff images were first subject to 'Gaussian blur' (Sigma = 1 pixel), 'subtract background' (rolling ball based, r = 20 pixels) and 'threshold' (manually adjusted parameters to retain most comets). Effective area (excluding nuclear area) was obtained by subjecting images to 'threshold' so that only the cytoplasmic region was highlighted. EB1 comet density was calculated by dividing the number of EB1 comets by the effective area.

Quantifications of GIP-2::GFP intensity were performed automatically using an imageJ (FIJI) macro script. For measuring the intensity of GIP-2::GFP puncta in the epidermis, tiff images were first subjected to 'Gaussian blur' (Sigma = 10 pixels) and 'setThreshold' (lowThreshold = 0, highThreshold = 205) to get the total imaged area (Image Area). The images were then subjected to 'Gaussian blur' (Sigma = 1 pixel) and 'Find Maxima...' (noise = 2) to pinpoint all GFP puncta. GFP puncta were enclosed for signal measurements by expanding the selection by 3 pixels on all sides to generate 7-pixel diameter circular regions centered over the 1-pixel-sized selections; area [in] and mean intensity [in] were measured for these regions. Background measurements were generated for each region by expanding the 7-pixel diameter circles by 3 more pixels on each side to generate 13-pixel diameter circular regions centered over the same 1-pixel-sized selections and measuring Area [out] and Mean Intensity [out]. The average background intensity was calculated as Background Intensity = (Area [out] × Mean Intensity [out] − Area [in] × Mean Intensity [in])/(Area [out] − Area [in]). The total intensity of the signal was calculated as Total Intensity = Area [in] × (Mean Intensity [in] − Background Intensity). The normalized intensity is the Total Intensity divided by the Image Area. For measuring the membrane and centrosomal GIP-2::GFP intensity in the germline, the Image Area was obtained using exactly the same strategy as in the epidermis. The tiff images were subjected to 'Gaussian blur' (sigma = 1 pixel) and 'setThreshold' (lowThreshold = 0, highThreshold = 211) to narrow down the selection to membranes and centrosomes. To get all signals, the selection was expanded by adding 6 pixels on all sides and measuring Area [in] and Mean Intensity [in] in this region. The background was calculated by expanding the selected area by 6 more pixels on all sides to measure Area [out] and Mean Intensity [out]. The Background Intensity, Total Intensity, and normalized intensity were then calculated as described above for the epidermis.

## Antibody production

To generate the NOCA-1 and PTRN-1 antibodies, the regions encoding amino acids 569–717 of NOCA-1h and 910–1110 of PTRN-1a were amplified from an N2 cDNA library using the oligos listed in *Table 3* and cloned into pGEX6P-1. GST fusions were purified from bacteria and outsourced for injection into rabbits (Covance, Princeton, NJ). NOCA-1 antibodies were affinity purified using the same antigen after cleavage of the GST tag as previously described (*Desai et al., 2003*). PTRN-1 antibodies were affinity purified using a GFP-PTRN-1-6×His fusion purified from baculovirus-infected insect cells (see 'Protein purification' section).

**Table 3**. Oligos used in antibody production

| Target | Oligonucleotide 1 | Oligonucleotide 2 | Template |
|--------|-------------------|-------------------|----------|
| NOCA-1 | ttgaattcCTCcgattgcaagaaatga | ttgaattcTTAgagttcttcaactgctcg | N2 cDNA |
| PTRN-1 | aagttctgttccaggggccc AAGGAGCTCGGTGCTGAG | agtcgacccgggaattctta GTTATTCTTATGAGCCGGAGTTC | N2 cDNA |

## Western blots

For the Western blot in *Figure 1B* right panel, *Figure 4E* and *Figure 1—figure supplement 2*, 20–50 control, *noca-1(RNAi)* or *noca-1h(RNAi)* worms were transferred into a pre-weighed Eppendorf tube containing 1 ml of M9 + 0.1% Triton X-100 and washed three times. After the last wash, excess buffer was removed until the net weight of worms and buffer was proportional to the number of worms (1 mg per worm) and 1/3 vol of 4× sample buffer was added. Worms were then placed in a sonicating water bath at ~70°C for 10 min and boiled for 3 min. For all other worm Western blots, a mixed population of worms growing at 20°C were collected and washed three times in M9 + 0.1% Triton X-100 in an Eppendorf tube. After the last wash, ~100 µl buffer was left, and then 50 µl of 4× sample buffer and 100 µl of glass beads (Sigma–Aldrich; G8772) were added. Worms were vortexed for 5 min, boiled for 3 min, and then vortexed and boiled again.

Samples were separated on an SDS-PAGE gel, transferred to a nitrocellulose membrane, probed with 1 µg/ml anti-NOCA-1, anti-PTRN-1 or anti-α-tubulin (mouse monoclonal DM1-α; Sigma–Aldrich, St. Louis, MO; T9026), and detected with an HRP-conjugated secondary antibody (rabbit or mouse; GE Healthcare, Little Chalfont, United Kingdom).

## Microtubule co-sedimentation from worm extract

The *C. elegans* extract was made as previously described (*Zanin et al., 2011*). Briefly, ~1 g of frozen worms from a large-scale liquid culture were weighed and resuspended in 1.5× vol of worm lysis buffer (50 mM Hepes-KOH pH 7.6, 1 mM $MgCl_2$, 1 mM EGTA, 75 mM KCl, 0.5 mM DTT, 1 µg/ml Pepstatin A, 1 tablet of Roche cOmplete EDTA-free protease inhibitor per 50 ml). The suspension was sonicated to obtain a crude extract. The crude extract was centrifuged at 20,000×g in a TLA100.3 rotor (Beckman, Pasadena, CA) for 10 min at 2°C, and the supernatant was re-centrifuged at 50,000×g using a TLA100.3 rotor (Beckman) for 20 min at 2°C. The supernatant after the second centrifugation was used for the experiment.

To determine whether NOCA-1 and PTRN-1 co-pellet with taxol-stabilized microtubules, a procedure modified from *Kellogg et al. (1989)* was used. For each experimental condition, 200 µl of worm extract prepared as above was supplemented with 0.5 mM DTT, 1 mM GTP, and 0.4 µl of DMSO (solvent control), 0.5 mg/ml nocodazole (no microtubule control) or 10 mM taxol (stabilized microtubule). The samples were warmed to room temperature (~23°C) for 10 min to allow microtubule polymerization, incubated on ice for 15 min and then pelleted through a glycerol cushion (worm lysis buffer with 40% glycerol) by centrifugation at 48,000×g in a TLA120.2 rotor (Beckman) for 30 min at 2°C. The sample/cushion interface was washed three times with worm lysis buffer. The pellet was resuspended in 200 µl of 1× sample buffer, and 12 µl of each sample were separated on an SDS-PAGE gel for either Coomassie staining or Western blots.

## Protein purifications

For microtubule anchoring and gliding assays, DmPatronin, PTRN-1, NOCA-1[LICR+NHD], NOCA-1[NHD], and MBP-NOCA-1[NHD] were PCR-amplified from a plasmid or N2 cDNA, and then cloned into the pFL vector (*Berger et al., 2004*) downstream of the p10 viral promoter with GFP and His tags. Plasmids were transformed into DH10EMBacY to generate bacmids, which were transfected into Sf9 cells to produce baculovirus. High Five or Sf9 cells were infected at $1–2 \times 10^6$ cells/ml using the baculoviruses (1:50 or 1:100 dilutions for High Five and 1:10 dilutions for Sf9 cells) and cultured for 48 hr at 27°C (High Five cells) or 24.5°C (Sf9 cells) before being collected. Expression of GFP-tagged protein was monitored using a fluorescence dissection scope.

For purifications of DmPatronin, PTRN-1, NOCA-1[LICR+NHD], and NOCA-1[NHD] with GFP and 6×His tags, baculovirus-infected High Five cells from 150 ml culture were lysed by sonication in 50 ml lysis buffer (50 mM Hepes pH 7.6, 500 mM NaCl, 10 mM imidazole, 10% sucrose, 1 mM DTT, 0.1% Tween-20, 1 µg/ml pepstatin A, 1 tablet of Roche cOmplete EDTA-free protease inhibitor). The crude extract was spun at 40,000 rpm in a Ti 45 rotor (Beckman) for 30 min at 4˚C, and the soluble fraction was incubated with 1-ml nickel beads for 1 hr at 4˚C. The beads were then washed three times with 50 ml wash buffer (25 mM Hepes pH 7.6, 500 mM NaCl, 25 mM imidazole, 10% sucrose, 1 mM DTT, 0.1% Tween-20) and eluted with 1 ml fractions of elution buffer (25 mM Hepes pH 7.6, 500 mM NaCl, 250 mM imidazole, 10% sucrose, 1 mM DTT, 0.1% Tween-20). Elutions were either used for flow-cell assays directly or were stored at −80˚C after snap-freezing of 50–100 µl aliquots in liquid nitrogen.

For purifications of MBP::NOCA-1[NHD]::GFP-6×His and MBP::GFP::6×His, baculovirus-infected Sf9 cells from 150 ml culture were lysed by sonication in 50 ml lysis buffer (50 mM Hepes pH 7.6, 500 mM NaCl, 10% glycerol, 1 mM DTT, 0.1% Tween-20, 1 µg/ml pepstatin A, 1 tablet of Roche cOmplete EDTA-free protease inhibitor). The crude extract was spun at 40,000 rpm in a Ti 45 rotor (Beckman) for 30 min at 4˚C, and the soluble fraction was incubated with 1 ml nickel beads for 1 hr at 4˚C. The beads were then washed three times with 50 ml wash buffer (25 mM Hepes pH 7.6, 500 mM NaCl, 10% glycerol, 1 mM DTT, 0.1% Tween-20) and eluted with 1 ml fractions of elution buffer (25 mM Hepes pH 7.6, 500 mM NaCl, 10% glycerol, 1 mM DTT, 0.1% Tween-20, 10 mM maltose). Elutions were either used for flow-cell assays directly or dialyzed into the dialysis buffer (25 mM Hepes pH 7.6, 500 mM NaCl, 10% glycerol, 1 mM DTT) and then used for flow-cell assays.

The kinesin motor domain (K560; Woehlke et al., 1997) with or without GFP tag was expressed in Escherichia coli cells (Rosetta or BL21) induced at OD600 0.6–0.8 and cultured at 13˚C overnight. Bacteria from 1.5 l culture were lysed in 50 ml lysis buffer (50 mM Hepes-K pH 7.6, 500 mM KCl, 10 mM imidazole, 10% Glycerol, 1 mM DTT, 1 mM ATP, 1 mM MgCl$_2$, 1 µg/ml pepstatin A, 1 tablet of Roche cOmplete EDTA-free protease inhibitor). The crude extract was spun at 40,000 rpm in a Ti 45 rotor (Beckman) and the soluble fraction was incubated with 1 ml nickel beads for 1 hr at 4˚C. The beads were then washed three times with 50 ml wash buffer (50 mM Hepes-K pH 7.6, 500 mM KCl, 25 mM imidazole, 10% Glycerol, 1 mM DTT, 1 mM ATP, 1 mM MgCl$_2$) and eluted with 1 ml fractions of elution buffer (50 mM Hepes-K pH 7.6, 500 mM KCl, 250 mM imidazole, 10% Glycerol, 1 mM DTT, 1 mM ATP, 1 mM MgCl$_2$). Elutions were used for flow-cell assays either directly or stored at −80˚C after snap-frozen in 50–100 µl aliquots in liquid nitrogen.

## Microtubule flow-cell assays

Coverslips were cleaned by sonication for 10 min in 5 M KOH dissolved in pure ethanol followed by 10 min of sonication in clean water, 2× rinse with water and 1× rinse with ethanol. After being dried in 37˚C incubator for overnight, the coverslips were used to make flow cells using microscope slides (Gold Seal; Thermo Scientific, Waltham, MA) and double-sided tape (Scotch, St. Paul, MN).

To make rhodamine-labeled GMPCPP microtubules, labeled and unlabeled bovine tubulin were clarified by centrifugation at 90,000 rpm using a TLA120.2 rotor (Beckman) for 5 min at 2˚C. Then the concentrations of labeled and unlabeled tubulins were measured. An elongation mix was prepared by mixing labeled and unlabeled bovine tubulin at a 1:20 ratio and 10 µM total concentration in BRB80 (80 mM Pipes-KOH pH 6.8, 1 mM MgCl$_2$, 1 mM EGTA) supplemented with 1 mM DTT and 0.5 mM GMPCPP (NU-405; Jena Bioscience, Jena, Germany). The mix was snap frozen in liquid nitrogen at 5 µl and stored at −80˚C. The stock of labeled microtubules were made by thawing an elongation mix aliquot, diluting with 5 µl of BRB80 plus 1 mM DTT and incubating in a 37˚C water bath for 30–60 min.

For microtubule anchoring assays, 10 µg/ml of llama GFP nanobody diluted in Tris-KAc buffer (50 mM Tris-HCl 8.0, 150 mM KAc, 10% Glycerol, 1 mM DTT) was introduced into the flow cell and incubated for 5 min. Then the coverslip was blocked by flowing in 1 mg/ml casein diluted in Tris-KAc buffer and incubating for 5 min. 5 nM of the GFP fusion to be tested diluted in Tris-KAc buffer were flowed in and incubated for another 5 min. Finally, 0.1 µM of rhodamine-labeled GMPCPP microtubules diluted in microtubule buffer with an oxygen scavenger mix (1×BRB80, 1 mM DTT, 1 mg/ml casein, 0.8 mg/ml glucose, 0.04 mg/ml glucose oxidase, 0.016 mg/ml catalase) was flowed in before imaging.

For kinesin gliding assays, the kinesin motor domain K560 (most concentrated fraction after His purification) was introduced into the flow cell and incubated for 5 min. The coverslip was blocked with the Gliding Buffer (1×BRB80, 1 mg/ml casein, 100 mM KCl, 0.1% Tween-20, 10% sucrose, 1 mM DTT, 1 mM

ATP) and 0.1 µM of rhodamine-labeled GMPCPP microtubules diluted in Gliding Buffer was flowed in and incubated for 5 min. Finally, samples containing ~200 pM of GFP-DmPatronin or ~300 pM of GFP-PTRN-1 diluted in the Gliding Buffer, 60 nM of MBP-NOCA-1$^{NHD}$-GFP or MBP-GFP diluted in BRB80 with 1 mg/ml Casein or 1 µM of MBP-NOCA-1$^{NHD}$-GFP diluted in H100 (25 mM Hepes-NaOH pH 7.6, 100 mM NaCl and 1 mM DTT) with 1 mg/ml Casein (all supplemented with the oxygen scavenger mix: 0.8 mg/ml glucose, 0.04 mg/ml glucose oxidase, 0.016 mg/ml catalase) were flowed in before imaging.

## Microtubule co-sedimentation assay

To make taxol stabilized microtubules, 2 mg/ml bovine tubulin in BRB80 with 1 mM DTT and 1 mM GTP was clarified by centrifugation at 90 k rpm using a TLA120.2 rotor (Beckman) for 5 min at 2°C. The clarified tubulin was incubated at 37°C for 2 min, and then 2 µM, 20 µM and 200 µM taxol was added stepwise. Each taxol addition was followed by 10 min of incubation at 37°C. Polymerized microtubules were then pelleted through a 40% glycerol in BRB80 cushion in a pre-warmed TLA120.2 rotor at 80 k rpm for 20 min at 25°C. The pellet was resuspended in BRB80 with 200 µM taxol, and the concentration was determined by the absorbance at 280 nm.

For microtubule co-sedimentation assay, reaction mixes of 1 µl of 10 mM taxol in DMSO, 5 µl of 60 µM microtubules or microtubule resuspension buffer, 74 µl of H100 buffer (25 mM Hepes-NaOH pH 7.6, 100 mM NaCl, and 1 mM DTT) and 20 µl test protein in dialysis buffer (25 mM Hepes-NaOH pH 7.6, 100 mM NaCl, 1 mM DTT and 10% Glycerol) or dialysis buffer alone were incubated at room temperature for 5 min. 90 µl of each reaction mix was layered onto 100 µl of cushion (40% glycerol in 50 mM Hepes-KOH pH 7.6, 75 mM KCl, 1 mM MgCl$_2$, 1 mM EGTA, 1 mM DTT, and 3 µM taxol) in a tiny centrifuge tube and spun at 100 k rpm for 10 min at 25°C using a TLA100 rotor (Beckman). 30 µl of supernatant sample was taken from the top of each tube, and 10 µl of 4× sample buffer was added in each sample. The cushion–layer interface was subsequently washed for three times using the H100 buffer before all supernatant was removed. The pellet was then re-suspended in 120 µl of 1× sample buffer. Supernatant and pellet samples were separated on an 8% poly-acrylamide gel for Coomassie blue staining.

## Acknowledgements

We thank K Laband for assistance in making the NOCA-1 antibody; R Khaliullin for help with python scripts and image analysis; N Hattersley for providing the purified GFP nanobody; S Dehenau for help with the palmitoylation prediction; D Starr for NOCA-1h identification; H Fridolfsson for nuclear migration assay training; JS Han, W Lan, and DH Kim for providing insect cells and baculovirus support; M Chuang, S Xu, A Chisholm, M Hendershott, and R Vale for plasmids; and A Chisholm and OD lab members for helpful discussions. The Caenorhabditis Genetics Center and NemaGENETAG provided strains. KO and AD receive salary and other support from the Ludwig Institute for Cancer Research. This work was supported by an NIH grant (GM074207) to KO.

## Additional information

### Funding

| Funder | Grant reference | Author |
| --- | --- | --- |
| National Institutes of Health | GM074207 | Karen Oegema |

The funder had no role in study design, data collection and interpretation, or the decision to submit the work for publication.

### Author contributions

SW, Conception and design, Acquisition of data, Analysis and interpretation of data, Drafting or revising the article, Contributed unpublished essential data or reagents; DW, Acquisition of data, Analysis and interpretation of data, Drafting or revising the article; SQ, Conception and design, Drafting or revising the article, Contributed unpublished essential data or reagents; RAG, Conception and design, Analysis and interpretation of data, Drafting or revising the article, Contributed unpublished essential data or reagents; DKC, SDO, Drafting or revising the article, Contributed unpublished essential data or reagents; AD, KO, Conception and design, Analysis and interpretation of data, Drafting or revising the article

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
