## [Decision Letter]

Thank you for submitting your work entitled “The *C. elegans* Ninein-Related Protein NOCA-1 Binds Microtubule Ends and Organizes Non-Centrosomal Microtubule Arrays” for peer review at *eLife*. Your submission has been favorably evaluated by Fiona Watt (Senior Editor) and three reviewers, one of whom is a member of our Board of Reviewing Editors.

The reviewers have discussed the reviews with one another and the Reviewing editor has drafted this decision to help you prepare a revised submission.

This manuscript identifies a putative *C. elegans* ortholog of ninein (called NOCA-1), characterizes its in vivo functional roles and its microtubule binding properties and demonstrates genetic interactions with the minus-end binding protein Ptrn-1/Patronin. NOCA-1 was previously shown by the Oegema group to be required for germline development and fertility. Here, they find that it genetically interacts with and is synthetic lethal with mutations in Ptrn-1 (a minus end binding protein). The essential function occurs in the *epidermis* where NOCA-1 and Ptrn-1 collaborate to generate bundled microtubule arrays. The authors perform detailed studies to determine the isoforms required for function of NOCA-1 in three distinct tissues. Furthermore, the authors purify NOCA-1 fragments to get at the underlying mechanism/biochemical properties of NOCA-1 and conclude that it can bind to microtubule ends, although, in contrast to in vivo data which point to microtubule minus-end related function, a bias for microtubule plus-end binding is observed.

Essential revisions:

1) The conclusion about NOCA-1 binding to microtubule ends is mostly based on one experiment, which is not very convincing. First, the control experiment with kinesin does not provide a fair comparison, because the kinesin (K560-GFP) is abundantly absorbed on glass, while NOCA-1 and Patronin are present as very sparse dots. It also appears that Patronin and NOCA-1 are present in clusters, while the kinesin is not. It is possible that if larger kinesin clusters were sparsely attached to glass, binding to microtubule ends would be observed as well. The control thus needs to be made more comparable to the “difficult” NOCA-1 preparation. Second, since Patronin does not show preference for the minus ends in this assay, the specificity of the assay is uncertain. Therefore, the authors should strongly downplay their conclusions on NOCA-1 being a microtubule end-binding protein, especially in the title and Abstract. Alternatively, they should provide additional data supporting specific microtubule minus end binding. Since the possibilities to work with NOCA-1 in vitro are strongly limited by its poor solubility, perhaps the authors can perform some experiments with GFP-NOCA-1 in cells (such as mammalian cells or any other cellular model) to prove that NOCA-1 indeed binds to microtubule ends.

2) The data supporting the conclusion on NOCA-1 organizing non-centrosomal microtubule arrays are quite weak. There is a clear loss of non-centrosomal arrays in the *epidermis* upon loss of NOCA-1 and Ptrn-1, though the actual functions of each one are still unclear (and the arrays are fine upon loss of NOCA-1 alone). In the other tissues, no data was given to show that non-centrosomal microtubule arrays were in fact disrupted and that NOCA-1 wasn't required for the proper dynamics of these structures rather than their formation and or organization. Thus, this seems to be based largely on the way that others have thought about ninein function in mammalian cells. It is also unclear how the minus end binding activity results in the phenotypes seen in vivo (such as changes in microtubule growth rates. The conclusions, especially in the title and Abstract of the paper, must be formulated more cautiously, and a better discussion should be provided.

3) In Figure 1, the authors document distal gonad defects, with γ-tubulin and *noca-1* mutants being similar. While the authors quantify brood sizes, they do not indicate whether the distal gonad cellular defects are reproducible. Some quantification of these defects would improve the manuscript.

4) The homology with ninein seems to be rather weak. Is it more significant than similarity to any other coiled coil protein(s)? In the alignment in the Figure 1—figure supplement 1, it would be helpful to indicate the positions of hydrophobic residues in the heptad repeat, so that it becomes more clear which additional residues are conserved. If the ninein homology is borderline, the point about NOCA-1 being a “ninein-related protein” would need to be downplayed, especially in the title and Abstract of the paper.

5) The authors suggest that cell surface targeting of NOCA-1h occurs through the palmitoylation site in NOCA-1h specific region; although not essential, testing this prediction by mutating the site would strengthen the manuscript.

6) In the subsection “NOCA-1 and Patronin/PTRN-1 control assembly of a circumferential microtubule array required for the integrity of the larval/adult *epidermis*”, the authors ask if the microtubule array in the *epidermis* is important for morphogenesis, but fail to reference the landmark work of Priess and Hirsh (1987, Dev Biol) that includes evidence in support of such a role for microtubules. Indeed, the authors never reference this key publication with respect to the morphogenesis defects, which is surprising.

---

## [Author Response]

Prior to describing the significant revisions made in response to the reviewers’ comments, we would like to briefly summarize an entirely new approach that we developed and employed that has significantly extended our conclusions and is included in the revised manuscript. Both the approach and the results we obtained are shown in an entirely new Figure 3. In brief, our work had revealed parallel roles for NOCA-1 and PTRN-1 in the formation of a functional circumferential microtubule array in the larval *epidermis* of *C. elegans*. An important question raised by this finding was whether the γ-tubulin complex acted together with one or both of these factors in non-centrosomal array formation. This experiment was not feasible because the essential role of the γ-tubulin complex in cell division had precluded our ability to analyze its function in the larval *epidermis*. We were able to overcome this limitation by developing a strategy that enabled selective removal of an essential subunit of the γ-tubulin small complex, GIP-2, in the larval *epidermis*. In this strategy, we first tagged the endogenous locus encoding GIP-2 with GFP and confirmed functionality of the GIP-2::GFP. Next we engineered a GFP degron that can act in a tissue-specific manner (our degron strategy was inspired by two prior approaches, as described in the manuscript text). We demonstrated that this approach specifically degraded GIP-2::GFP in the larval *epidermis*, and thus allowed us to test γ-tubulin complex function in that tissue. By combining the GIP-2::GFP and the tissue-specific GFP degron with *noca-1∆* or *ptrn-1∆*, we obtained a very clear result – the γ-tubulin complex acts together with NOCA-1 (i.e. degradation of GIP-2 does not enhance *noca-1∆)* and in parallel to PTRN-1 (i.e. degradation of GIP-2 strongly enhances *ptrn-1∆*). We believe this is a very important result that significantly extends the conclusions presented in our study and should guide future analysis of various minus end-directed activities in other systems.

*Essential revisions*:

*1) The conclusion about NOCA-1 binding to microtubule ends is mostly based on one experiment, which is not very convincing. First, the control experiment with kinesin does not provide a fair comparison, because the kinesin (K560-GFP) is abundantly absorbed on glass, while NOCA-1 and Patronin are present as very sparse dots. It also appears that Patronin and NOCA-1 are present in clusters, while the kinesin is not. It is possible that if larger kinesin clusters were sparsely attached to glass, binding to microtubule ends would be observed as well. The control thus needs to be made more comparable to the “difficult” NOCA-1 preparation. Second, since Patronin does not show preference for the minus ends in this assay, the specificity of the assay is uncertain. Therefore, the authors should strongly downplay their conclusions on NOCA-1 being a microtubule end-binding protein, especially in the title and Abstract. Alternatively, they should provide additional data supporting specific microtubule minus end binding. Since the possibilities to work with NOCA-1* in vitro *are strongly limited by its poor solubility, perhaps the authors* can *perform some experiments with GFP-NOCA-1 in cells (such as mammalian cells or any other cellular model) to prove that NOCA-1 indeed binds to microtubule ends*.

Like PTRN-1, NOCA-1 co-sediments with microtubules from *C. elegans* extracts, suggesting that it binds, either directly or indirectly, to microtubules. As the reviewers highlight, the data supporting direct binding of NOCA-1 binding to microtubule ends has been limited by poor solubility of purified NOCA-1 fragments at physiological ionic strength. During the revision period, we did substantial additional work, outlined in detail below, aimed at clarifying the relationship between NOCA-1 clustering and microtubule binding. This effort revealed that we can only observe binding of purified NOCA-1 fusions to the microtubule lattice and/or microtubule ends when we induce the proteins to form small aggregates. As it is possible that this is an artifact that does not occur in vivo, we decided to *move the end anchoring assay data (old*
Figure 6*) to the supplement where we also now include the results of our new biochemical efforts*. Despite the fact that we do not yet have clarity on this issue, we think it is important to document what we have tried, which represents over a year of work, to provide a foundation for future efforts (the biochemistry data is now presented in two new figure supplements, Figure 1—figure supplement 4 and Figure 1—figure supplement 5). As suggested, we have removed the suggestion that NOCA-1 is a microtubule end-binding protein from the title and Abstract and have shifted the focus of the manuscript to the analysis of the relative roles of NOCA-1, PTRN-1 and γ-tubulin in the assembly of functional non-centrosomal microtubule arrays in three different *C. elegans* tissues. To facilitate this shift in focus, we added a new Figure 3, in which we employ a tissue-specific protein degradation system that we developed to deplete the essentialγ–tubulin complex component GIP-2 from the larval *epidermis*. This approach allowed us to build on our prior finding that NOCA-1 and PTRN-1 function in parallel pathways for microtubule generation in the larval *epidermis*, to show that theγ–tubulin complex promotes microtubule generation in the larval *epidermis* by acting with NOCA-1 and in parallel to the Patronin-dependent pathway. In accordance with these revisions the manuscript has been re-titled to: “NOCA-1 Functions withγ-tubulin and in Parallel to Patronin to Assemble Non-Centrosomal Microtubule Arrays in *C. elegans*”.

Summary of experiments to examine NOCA-1 microtubule binding:

In attempt to get more definitive data on the question of whether and how NOCA-1 binds to microtubules, we first attempted the approach suggested by the reviewers, and tried to use a polyclonal GFP antibody to cluster established microtubule side binding proteins (the kinesin K560-GFP) and the microtubule lattice binding protein (NDC80-GFP) to see if we could generate clusters that would anchor microtubule ends like we have seen for NOCA-1 and PTRN-1. This approach did not work, as we were not able to generate clusters of comparable size/morphology to those observed for NOCA-1 or Patronin. We also took the reviewers’ suggestion and tried to transiently expressed mRuby fused NOCA-1 in a U2OS cell line that stably expresses GFP labeled tubulin. However, the mRuby fusion of NOCA-1 formed large aggregates, preventing further analysis.

We were able to increase the solubility of the NOCA-1^NHD^-GFP fusion by addition of an MBP tag to its N-terminus. The resulting MBP-NOCA-1^NHD^-GFP was soluble in H100 buffer (25 mM Hepes-NaOH pH 7.6, 100 mM NaCl and 1 mM DTT) and eluted following gel filtration in a single major peak (new Figure 1—figure supplement 4) at a size that indicated that it was not aggregated. We performed microtubule co-sedimentation assays using 1.2 μM of the soluble MBP-NOCA-1^NHD^-GFP fusion and 3 μM of taxol-stabilized microtubules in H100 buffer, H50 buffer (25 mM Hepes-NaOH pH 7.6, 50 mM NaCl and 1 mM DTT) and HK75 buffer (50 mM Hepes-NaOH pH 7.6, 75 mM KCl, 1 mM MgCl_2_, 1 mM EGTA and 1 mM DTT), but did not observe microtubule co-pelleting (new Figure 1—figure supplement 4). Consistent with the co-sedimentation results, the soluble MBP-NOCA-1^NHD^-GFP fusion showed no microtubule binding at 1 μM in H100 buffer in a flow-cell based microtubule-gliding assay (new Figure 1—figure supplement 5). Interestingly, however, dilution of 6–60 nM MBP-NOCA-1^NHD^-GFP into the classic microtubule binding buffer BRB80 (80 mM Pipes-KOH pH 6.8, 1 mM MgCl_2_, 1 mM EGTA) induced the formation of small, homogeneous clusters that exhibited robust microtubule lattice binding (new Figure 1—figure supplement 5).

In summary, to date we have only observed binding of purified NOCA-1 proteins to microtubule lattice and/or ends when we induce the protein to form small aggregates. As it is possible that this is some type of artifact that does not occur in vivo, it remains unclear whether NOCA-1 can bind directly to microtubules. The biochemical analysis for purified NOCA-1 proteins is now summarized in two supplemental figures (Figure 1—figure supplement 4 and Figure 1—figure supplement 5) to guide future efforts.

*2) The data supporting the conclusion on NOCA-1 organizing non-centrosomal microtubule arrays are quite weak. There is a clear loss of non-centrosomal arrays in the epidermis upon loss of NOCA-1 and Ptrn-1, though the actual functions of each one are still unclear (and the arrays are fine upon loss of NOCA-1 alone). In the other tissues, no data was given to show that non-centrosomal microtubule arrays were in fact disrupted and that NOCA-1 wasn't required for the proper dynamics of these structures rather than their formation and or organization. Thus, this seems to be based largely on the way that others have thought about ninein function in mammalian cells. It is also unclear how the minus end binding activity results in the phenotypes seen* in vivo *(such as changes in microtubule growth rates. The conclusions, especially in the title and Abstract of the paper, must be formulated more cautiously, and a better discussion should be provided*.

We addressed this issue by adding images of the microtubule cytoskeleton (GFP::β-tubulin) for the germline and embryonic *epidermis* to the main figures (new panels in Figure 4 and Figure 6), along with additional explanation to the text to clarify the changes in each array and what has been assayed in each tissue context (discussed in detail below). To accommodate the additional data (and the reviewers request to partition out the isoform analysis), we split our original Figure 3, which had the data for both tissues, into two figures with the data for the germline array in the new Figure 4 (including new panels added as Figure 4) and for the embryonic epidermal array in the new Figure 6 (including new panels added as Figure 6). As discussed above, based on our re-assessment of whether NOCA-1 binds directly to microtubules and our new data analyzing the function of the γ-tubulin complex in the larval *epidermis*, we have changed the focus of the paper, along with the title and the Abstract. As requested, we have also provided a better discussion with a new accompanying model figure (new Figure 7).

Below we discuss the data from the different tissues and whether they suggest a role for NOCA-1 in microtubule organization vs microtubule assembly and/or dynamics. Note that in the previous version, we used “microtubule organization” in a general sense to imply that the microtubule arrays were functionally compromised when NOCA-1 was inhibited either alone (germline or embryonic *epidermis*) or in conjunction with PTRN-1 (larval *epidermis*). In the revised version we now use more specific language to describe the consequences of protein removal in each of the tissue contexts.

As the reviewers highlight, simultaneous inhibition of NOCA-1 and PTRN-1 leads to loss of microtubule bundles in the larval *epidermis*, indicating that at some level both of these proteins contribute to generating the microtubules that make up this array. The microtubule array in the larval *epidermis* is particularly suitable for analysis of the effect of protein inhibition on array structure and microtubule dynamics, since the overall structure of the tissue remains relatively constant upon depletion of one or both proteins, and we have now also been able to build a system to specifically knockdown an essentialγ–tubulin complex component after differentiation in this tissue (new Figure 3). Due to these experimental advantages, the larval *epidermis* is our “flagship” array that we describe in Figures 1, 2 and 3. In this array, inhibition of PTRN-1 alone does not appear to substantially alter microtubule dynamics, whereas NOCA-1 inhibition leads to a more stable microtubule array with substantially fewer growing microtubule ends that grow at a reduced rate (now shown in Figure 2 and Videos 2 and 3). Based on these results, we conclude that both NOCA-1 and PTRN-1 are involved in the generation of microtubules in the larval epidermal array and that the presence of NOCA-1 significantly alters the dynamics of microtubules within this array.

In the syncytial germline, we focused on the non-centrosomal arrays that form within the compartments in the pachytene region of the germline. For clarity, we have added schematics and immunofluorescence images of this array in control worms (new panels in Figure 4). These arrays hold the meiotic nuclei in juxtaposition to the cell surface within the compartments, which are open on one side to a cytoplasmic channel called the rachis that runs down the center of the germline, to prevent them from falling into the rachis. Within the rachis there are also a large number of microtubules that flow with the streaming cytoplasm into the forming oocytes. How these microtubules are related to the arrays within the compartments is not entirely clear. When we delete NOCA-1 or deplete γ-tubulin, all of the nuclei fall out of their compartments and cluster in the rachis, indicating a dramatic failure in the function of the compartment microtubule arrays. This nuclear fallout is coupled to degeneration of compartment structure (see Figure 4 and new Figure 4). Thus, unlike the situation in the larval *epidermis*, the fact that compartments with normal structure cease to exist as the microtubule arrays fail and the nuclei fall into the rachis, makes it impossible to do a proper “before and after” comparison to assess the impact of loss of NOCA-1 and γ-tubulin on the structure of the compartment arrays and the dynamics of their constituent microtubules. As a proxy, we examine the effect on the number of growing microtubule ends in the rachis (the only structure that exists before and after the inhibitions) and we observe that comet density is reduced ∼ 2-fold by NOCA-1 or γ-tubulin inhibition (Figure 4). From this, we can conclude that NOCA-1 and γ-tubulin are required to assemble/maintain functional microtubule arrays in the germline compartments. We now discuss the limitations of this tissue with respect to analyzing changes in microtubule organization and dynamics within the compartment arrays in the text.

In the embryonic *epidermis*, we have added new images to show the microtubule skeleton in control and *noca-1Δ* embryos (new panels added as Figure 6). These images revealed an apparent reduction in the intensity of the microtubule cytoskeleton in the absence of NOCA-1 – suggesting that there are fewer microtubules in these arrays. Consistent with this, we also observed a 2-fold reduction in the number of growing microtubule ends marked by EB1::GFP.

Cumulatively, these data suggest that NOCA-1 functions with γ-tubulin to promote microtubule assembly, which we now highlight in a model figure added to accompany the expanded discussion (new Figure 7). Our thinking about the functions of NOCA-1 was not that heavily impacted by the microtubule anchoring function that has been subscribed to ninein, as we did not realize NOCA-1 was a potential ninein homolog until we began to write up the work. However, we cannot exclude the possibility that NOCA-1 could have a role in microtubule anchoring in addition to its role in promoting assembly.

*3) In*
Figure 1*, the authors document distal gonad defects, with γ-tubulin and* noca-1 *mutants being similar. While the authors quantify brood sizes, they do not indicate whether the distal gonad cellular defects are reproducible. Some quantification of these defects would improve the manuscript*.

Thank you for picking up on this, which we overlooked in the original version due to the fact that the phenotypes are 100% penetrant and look the same in all worms. We now provide quantification of all of the phenotypes assessed by imaging. For clarity, the region of the gonad that we analyze is the region immediately prior to the turn, where nuclei in the pachytene stage of meiosis are held in their compartments by non-centrosomal arrays (rather than the distal region of the gonad where the nuclei undergo mitotic divisions). We show that *noca-1* deletion results in the same phenotype as depleting γ-tubulin by RNAi; nuclei fall out of their compartments and clump together in the rachis center. Deletion of *ptrn-1* does not cause this phenotype. We document this phenotype in two strains expressing either a GFP-tagged plasma membrane probe with mCherry::histone (now in Figure 4) or GFP::β-tubulin (moved from the supplement to Figure 4). We now include the number of germlines imaged and the penetrance of the nuclear fallout phenotype in the figure legend. Between the two strains we have imaged 36 control germlines (0% exhibit nuclear fallout), 17 *γ-tubulin(RNAi)* germlines (100% exhibit nuclear fallout), 25 *noca-1Δ* germlines (100% exhibit nuclear fallout) and 18 *ptrn-1Δ* germlines (0% exhibit nuclear fallout).

*4) The homology with ninein seems to be rather weak. Is it more significant than similarity to any other coiled coil protein(s)? In the alignment in the*
Figure 1—figure supplement 1*, it would be helpful to indicate the positions of hydrophobic residues in the heptad repeat, so that it becomes more clear which additional residues are conserved. If the ninein homology is borderline, the point about NOCA-1 being a “ninein-related protein” would need to be downplayed, especially in the title and Abstract of the paper*.

The homology between NOCA-1 and ninein is more significant than the similarity of either protein to other coiled coiled proteins. We have now added two new panels (new Figure 1—figure supplement 1) to make this point and highlight how the homology was identified. As we explain in the legend to this figure, the homology between nematode NOCA-1 homologs and vertebrate nineins was discovered in an NCBI blast using the conserved region of NOCA-1 from *Brugia malayi* as the query. In our experience, blasting using the *Brugia* sequence is one of the best ways to identify non-nematode homologs of *C. elegans* proteins, since *Brugia* sequences tend to be among the least divergent among nematode species. The blast using the *Brugia* sequence identified all of the nematode NOCA-1 homologs (new Figure 1—figure supplement 1, red, pink, and green text), along with the vertebrate *Chinchilla lanigera*, *Heterocephalus glaber* and *Fukomys damrensis* nineins (Figure 1—figure supplement 1, black and blue text), with no other significant hits. Importantly, reverse blast of aa 1549–1801 of *Chinchilla lanigera* ninein isoform X4 against all nematode sequences (new Figure 1—figure supplement 1) yielded *Brugia malayi* NOCA-1 as the top hit (E value = 2e-07) and Loa loa NOCA-1 as the second hit (E value = 2e-04). Other coiled-coil proteins were also detected, but with substantially less significant E values (i.e. *Toxocara canis* myosin II, E value = 3.3; *C. Briggsae* HCP-2, E value = 6.1). These data suggest that NOCA-1 is indeed more similar to ninein than to other coiled coiled proteins. To highlight this, we also marked the residues in the “a” and “d” positions of the coiled-coil heptad repeats on the alignment in Figure 1—figure supplement 1, as requested.

*5) The authors suggest that cell surface targeting of NOCA-1h occurs through the palmitoylation site in NOCA-1h specific region; although not essential, testing this prediction by mutating the site would strengthen the manuscript*.

This was a great suggestion. To address this, we made a new transgene expressing a GFP fusion with the full length germline NOCA-1h isoform with a point mutation in its predicted N-terminal palmitoylation site (NOCA-1h^C10A^::GFP). Like its wild-type counterpart, NOCA-1h^C10A^::GFP fully rescued the germline function of NOCA-1. Interestingly, however, unlike the wild-type protein, its localization to compartment surfaces depended on γ-tubulin. In addition, whereas depletion of γ-tubulin lead to failure of NOCA-1h^C10A^::GFP targeting, depletion of α-tubulin to an extent that phenocopied γ-tubulin or NOCA-1 inhibition did not. These experiments, now included as new panels in Figure 5, suggest that NOCA-1 is recruited to cell surfaces via a γ-tubulin-dependent mechanism and further highlights the close relationship between NOCA-1and γ-tubulin.

*6) In the subsection “NOCA-1 and Patronin/PTRN-1 control assembly of a circumferential microtubule array required for the integrity of the larval/adult epidermis”, the authors ask if the microtubule array in the epidermis is important for morphogenesis, but fail to reference the landmark work of Priess and Hirsh (1987, Dev Biol) that includes evidence in support of such a role for microtubules. Indeed, the authors never reference this key publication with respect to the morphogenesis defects, which is surprising*.

Instead of [49], we had referenced another paper from the Priess lab that describes the epidermal microtubule arrays in the larval stages (Costa et al., Dev. Biol. 1997), which are the arrays that we show here are required for larval growth. The 1986 paper focuses on an epidermal microtubule array in the embryo that forms subsequent to the embryonic arrays we characterize that position nuclei. We didn’t cite this reference because we don’t analyze this array or elongation in this manuscript, although we have done some collaborative work with the Labousse lab on this topic that will be the topic of another publication. However, since the microtubule array described in the 1986 Preiss and Hirsh paper immediately precedes the arrays that form in larva, it is reasonable to cite it along with the later 1997 paper, and we have added this reference to the revision.